# On the incongruence of genotype-phenotype and fitness landscapes

**Malvika Srivastava**[1,2], **Joshua L. Payne**[1,2]*

**1** Institute of Integrative Biology, ETH Zurich, Zurich, Switzerland, **2** Swiss Institute of Bioinformatics, Lausanne, Switzerland

* joshua.payne@env.ethz.ch

## Abstract

The mapping from genotype to phenotype to fitness typically involves multiple nonlinearities that can transform the effects of mutations. For example, mutations may contribute additively to a phenotype, but their effects on fitness may combine non-additively because selection favors a low or intermediate value of that phenotype. This can cause incongruence between the topographical properties of a fitness landscape and its underlying genotype-phenotype landscape. Yet, genotype-phenotype landscapes are often used as a proxy for fitness landscapes to study the dynamics and predictability of evolution. Here, we use theoretical models and empirical data on transcription factor-DNA interactions to systematically study the incongruence of genotype-phenotype and fitness landscapes when selection favors a low or intermediate phenotypic value. Using the theoretical models, we prove a number of fundamental results. For example, selection for low or intermediate phenotypic values does not change simple sign epistasis into reciprocal sign epistasis, implying that genotype-phenotype landscapes with only simple sign epistasis motifs will always give rise to single-peaked fitness landscapes under such selection. More broadly, we show that such selection tends to create fitness landscapes that are more rugged than the underlying genotype-phenotype landscape, but this increased ruggedness typically does not frustrate adaptive evolution because the local adaptive peaks in the fitness landscape tend to be nearly as tall as the global peak. Many of these results carry forward to the empirical genotype-phenotype landscapes, which may help to explain why low- and intermediate-affinity transcription factor-DNA interactions are so prevalent in eukaryotic gene regulation.

**Data Availability Statement:** All data used in this study are publicly available. See Aguilar-Rodriguez et al. (2017) "A thousand empirical adaptive landscapes and their navigability." Nature Ecology & Evolution, 1, 0045. Code used in this study is

## Author summary

How do mutations change phenotypic traits and organismal fitness? This question is often addressed in the context of a classic metaphor of evolutionary theory—the fitness landscape. A fitness landscape is akin to a physical landscape, in which genotypes define spatial coordinates, and fitness defines the elevation of each coordinate. Evolution then acts like a hill-climbing process, in which populations ascend fitness peaks as a consequence of mutation and selection. It is becoming increasingly common to construct such landscapes using experimental data from high-throughput sequencing technologies and

available at https://github.com/MSri95/
incongruence-of-gp-landscapes.

**Funding:** This work was funded by Swiss National
Science Foundation grants PP00P3_170604 and
310030_192541 to J.L.P. (https://www.snf.ch).
The funders had no role in study design, data
collection and analysis, decision to publish, or
preparation of the manuscript.

**Competing interests:** The authors have declared
that no competing interests exist.

phenotypic assays, in systems such as macromolecules and gene regulatory circuits.
Although these landscapes are typically defined by molecular phenotypes, and are there-
fore more appropriately referred to as genotype-phenotype landscapes, they are often
used to study evolutionary dynamics. This requires the assumption that the molecular
phenotype is a reasonable proxy for fitness, which need not be the case. For example,
selection may favor a low or intermediate phenotypic value, causing incongruence
between a fitness landscape and its underlying genotype-phenotype landscape. Here, we
study such incongruence using a diversity of theoretical models and experimental data
from gene regulatory systems. We regularly find incongruence, in that fitness landscapes
tend to comprise more peaks than their underlying genotype-phenotype landscapes.
However, using evolutionary simulations, we show that this increased ruggedness need
not impede adaptation.

## Introduction

Characterizing the relationship between genotype and phenotype is key to our understanding
of evolution [1, 2]. For quantitative phenotypes, such as the expression level of a gene or the
enzymatic activity of a protein, this relationship can be formalized as a genotype-phenotype
landscape [3]. In such a landscape, genotypes represent coordinates in an abstract genotype
space and their phenotype defines the elevation of each coordinate in this space [4]. The topo-
graphical properties of genotype-phenotype landscapes, such as their ruggedness, are influ-
enced by epistasis [5]—non-additive interactions between mutations in their contribution to
phenotype. These topographical properties have important evolutionary consequences,
because they determine how mutation brings forth the phenotypic variation upon which selec-
tion acts [6, 7].

Technological advances are facilitating the construction and analysis of empirical geno-
type-phenotype landscapes at ever-increasing resolution, scale, and scope [8]. Example pheno-
types include the enzymatic activity [9], binding affinity [10], allosteric profile [11], and
fluorescence intensity of proteins [12], as well as exon inclusion levels [3], the expression levels
of genes driven by regulatory elements [13], the expression patterns of gene regulatory circuits
[14–16], and flux through metabolic pathways [17]. In analyzing the topographical properties
of these landscapes and their evolutionary consequences, an assumption is often made that
phenotype is a proxy for fitness [12, 18–24], thus rendering genotype-phenotype landscapes
equivalent to fitness landscapes [25]. While this assumption may be justified under certain
conditions, such as in directed protein evolution experiments [26], it is often the case that the
relationship between phenotype and fitness is not so straightforward. For example, fitness may
depend upon more phenotypes than those being assayed [27] or the relationship between phe-
notype and fitness may be inherently nonlinear, for example reflecting a tradeoff between the
costs and benefits associated with a phenotype [28]. In the latter case, selection may favor a
low or intermediate phenotypic value [29, 30]; e.g., an intermediate gene expression level, [31–
33], enzyme efficiency [34] or protein production rate or activity [35]. Such non-linearities are
a cause of epistasis [16, 36–38], and they can transform the effects of mutations as they map
onto phenotype and fitness [37], thus rendering the topographical properties of a fitness land-
scape qualitatively different from those of its underlying genotype-phenotype landscape (Fig
1A). While we do not doubt that workers in the field are well aware that the topographical
properties of a fitness landscape can differ from those of its underlying genotype-phenotype
landscape, a systematic study of these differences is lacking.

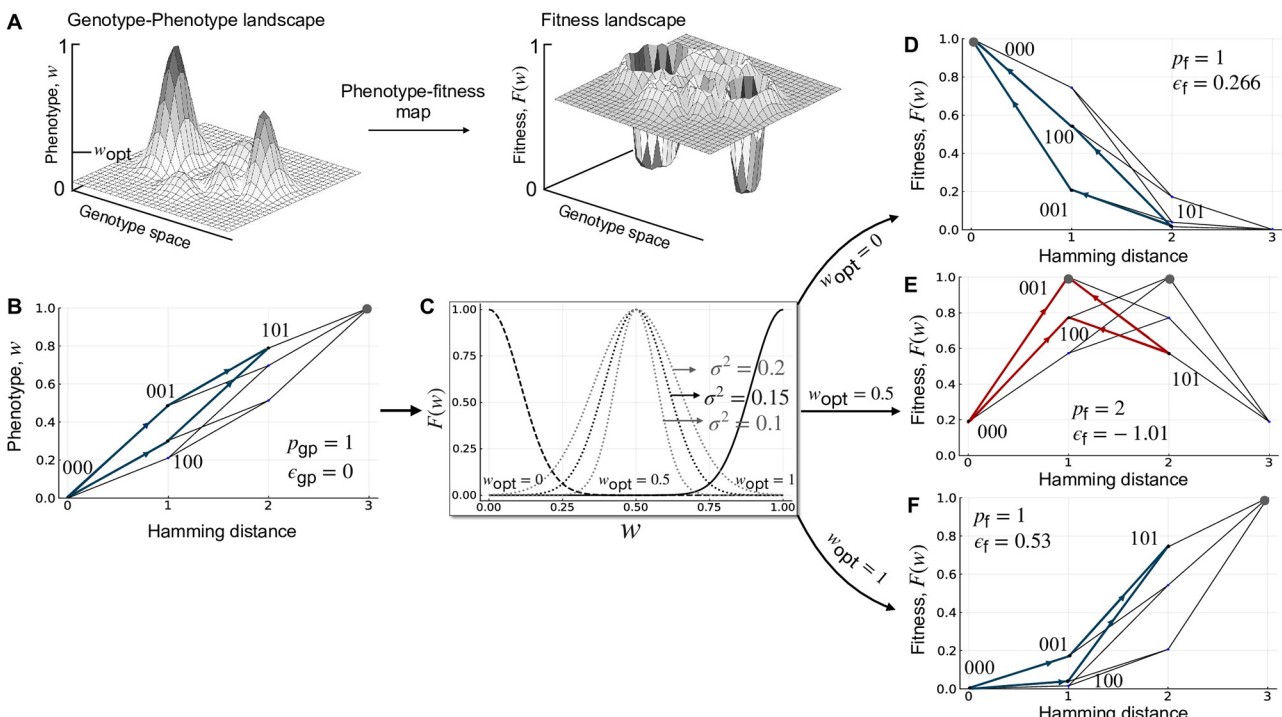

**Fig 1. Incongruence.** (A) Schematic illustration of how selection for an intermediate phenotypic value $w_{opt}$ can make a genotype-phenotype landscape incongruent with the resulting fitness landscape. (B) An additive three-locus, biallelic genotype-phenotype landscape with a single peak (gray filled circle; $p_{gp} = 1$). One pairwise interaction is highlighted in blue. It exhibits no magnitude epistasis ($\epsilon_{gp} = 0$) or sign epistasis. (C) The Gaussian phenotype-fitness map (Eq 1) is shown for three values of $w_{opt}$ (dashed line, $w_{opt} = 0$; dotted line, $w_{opt} = 0.5$; solid line, $w_{opt} = 1$), with three values of $\sigma$ shown for $w_{opt} = 0.5$. (D) Applying the Gaussian phenotype-fitness map with $w_{opt} = 0$ to the genotype-phenotype landscape results in a single-peaked fitness landscape (gray filled circle; $p_f = 1$). The same pairwise interaction from (B) is highlighted in blue. It exhibits positive epistasis ($\epsilon_{gp} = 0.266$), but no sign epistasis. (E) Applying the Gaussian phenotype-fitness map with $w_{opt} = 0.5$ to the genotype-phenotype landscape results in a multi-peaked fitness landscape (gray filled circles; $p_f = 2$). The same pairwise interaction from (B) is highlighted in red. It exhibits negative epistasis ($\epsilon_{gp} = -1.01$), as well as reciprocal sign epistasis. (F) Applying the Gaussian phenotype-fitness map with $w_{opt} = 1$ to the genotype-phenotype landscape results in a single-peaked fitness landscape (gray filled circle; $p_f = 1$). The same pairwise interaction from (A) is highlighted in blue. It exhibits positive epistasis ($\epsilon_{gp} = 0.53$), but no sign epistasis. In panels B, D-F, arrows point from genotypes with lower phenotypic or fitness values to genotypes with higher phenotypic or fitness values. The no sign epistasis motif is highlighted in blue and the reciprocal sign epistasis motif in red. Note the symmetry of landscapes in panels D and F.

Selection for low or intermediate phenotypic values is especially relevant to transcription factor-DNA interactions [39]. Transcription factors are sequence-specific DNA binding proteins that help regulate gene expression. They do so by binding DNA sequences (transcription factor binding sites) in regulatory regions such as promoters and enhancers to recruit or block the recruitment of RNA polymerase [40]. The regulatory effect of such a binding event depends in part on the affinity with which the DNA sequence is bound by the transcription factor [41, 42]. As such, binding affinity is an important molecular phenotype of transcription factor binding sites, upon which selection acts [43, 44]. While it is commonly assumed that selection increases binding affinity [21, 43–47], several lines of evidence suggest that selection for low or intermediate binding affinity also influences the evolution of transcription factor binding sites. For example, paralogous transcription factors often bind the same DNA sequences with high affinity, but different DNA sequences with low affinity [48, 49]. If an optimal gene expression pattern requires binding by just one of several transcription factor paralogs, such specificity can be achieved using low-affinity transcription factor binding sites, resulting in selection for low binding affinity [50]. Additional documented cases in which low-affinity binding sites play important regulatory roles include negative auto-regulation by

high-copy number transcription factors in *Escherichia coli* [51], where high-affinity binding sites cause suboptimal noise suppression, and developmental patterning in *Ciona intestinalis* embryos [52, 53], where high-affinity binding sites cause deleterious ectopic gene expression patterns. Moreover, low-affinity binding sites are commonly observed in the regulatory portfolios of a diversity of organisms, including bacteria [51], yeast [54], fly [49, 55, 56], sea stars and sea urchins [57], as well as humans [58].

How incongruent are the topographies of genotype-phenotype and fitness landscapes when selection favors a low or intermediate phenotypic value? How does this depend on the ruggedness of the genotype-phenotype landscape? These are important questions, because the topography of a fitness landscape has implications for several evolutionary phenomena, including the evolution of genetic diversity [59], reproductive isolation [60], and sex [61], as well as the predictability of the evolutionary process itself [62]. How much we can learn about these phenomena from knowledge of a genotype-phenotype landscape depends on the genotype-phenotype landscape's congruence with the fitness landscape. Despite decades of research on fitness landscapes [63, 64], these questions have not been addressed even in the context of classical theoretical models, such as Mt. Fuji [65], House-of-Cards [66], or NK landscapes [67]. They have also not been addressed in the context of biophysical models of genotype-phenotype landscapes or empirical genotype-phenotype landscapes. Here, we fill this knowledge gap by defining local and global measures of incongruence, which describe the topographical differences between a genotype-phenotype landscape and the corresponding fitness landscape when selection favors a low or intermediate phenotypic value. We use these measures to study incongruence in the context of the aforementioned theoretical models [65–67] and derive some fundamental results that are applicable to all empirical genotype-phenotype landscapes. We then consider the specific case of genotype-phenotype landscapes of transcription factor-DNA interactions, by first looking at an idealised biophysical model [68] and then taking a step further, by analysing 1,137 empirical genotype-phenotype landscapes, wherein genotypes are transcription factor binding sites and the phenotype is a measure of relative binding affinity [21]. We study transcription factor-DNA interactions because there is strong biological motivation for studying selection for low or intermediate binding affinity, as discussed above, and because a large number of empirical genotype-phenotype landscapes of transcription factor-DNA interactions are publicly available [21], thus facilitating the statistical analysis of their incongruence.

## Results

We first present our measures of incongruence and the phenotype-to-fitness map used to incorporate the effect of selection for a low or intermediate phenotypic value. We use these measures and this map to study the incongruence of randomly generated genotype-phenotype landscapes, specifically two-locus biallelic landscapes and multi-locus biallelic landscapes. For the latter, we use NK landscapes [67], which include the corner cases of Mt. Fuji [65] ($K = 0$) and House-Of-Cards [66] ($K = N - 1$) landscapes (see Table 1 for a list of symbols). We then apply the principles learned from these model genotype-phenotype landscapes to genotype-phenotype landscapes of transcription factor-DNA interactions, first in the context of the mismatch model [68] (which enables us to study landscapes with more than two alleles per locus), and then in the context of empirical measurements of transcription factor-DNA interactions [21]. Finally, we study the evolutionary consequences of landscape incongruence.

### Landscape incongruence

Our goal is to quantify the topographical differences between a fitness landscape and its underlying genotype-phenotype landscape, when selection favors a low or intermediate phenotypic

**Table 1. List of symbols.**

| | |
|---|---|
| $w$, $w_{opt}$ | Phenotypic value, Optimal phenotypic value |
| $\sigma$ | Strength of selection |
| $F(w)$ | Fitness value |
| $\epsilon_{gp}$, $\epsilon_f$ | Epistasis in the genotype-phenotype landscape (gp) and fitness landscape (f) respectively |
| $p_{gp}$, $p_f$ | Number of peaks in the genotype-phenotype landscape (gp) and fitness landscape (f) respectively |
| $L$ | Length of the sequence |
| $K$ | Ruggedness parameter of the NK model |
| $a$ | Number of alleles at each locus |
| $m$ | Number of mismatches in the mismatch model |
| $\langle l \rangle$ | Average length of adaptive walk |
| $\langle f \rangle$ | Mean fitness at equilibrium |

value. To do so, we define measures of landscape incongruence, at both a local and a global scale. At a local scale, we quantify differences in pairwise epistasis amongst loci in the genotype-phenotype landscape relative to the same loci in the fitness landscape (Methods). We do so by classifying the type of magnitude epistasis ($\epsilon$) as additive (i.e., no epistasis; $\epsilon = 0$), positive ($\epsilon > 0$), or negative ($\epsilon < 0$). For a pair of loci, we then compare the type of epistasis in the genotype-phenotype landscape ($\epsilon_{gp}$) to the type of epistasis in the fitness landscape ($\epsilon_f$), and report whether this is the same in the two landscapes. We do the same for an additional classification of epistasis based on the absence or presence of sign epistasis. This results in three categories— no sign epistasis, simple sign epistasis, or reciprocal sign epistasis [69]. At a global scale, we quantify incongruence as the difference in the number of peaks in the fitness landscape ($p_f$) relative to the genotype-phenotype landscape ($p_{gp}$). Taken together, these local and global measures allow us to determine the extent to which selection for a low or intermediate phenotypic value increases or decreases the ruggedness of the fitness landscape, relative to the genotype-phenotype landscape.

## Phenotype-to-fitness map

To study the effect of selection for low or intermediate phenotypic values, we use a Gaussian phenotype-to-fitness map $F(w)$ centred around an optimal phenotypic value $w_{opt}$,

$$F(w) = \exp\left[-\left(\frac{w - w_{opt}}{\sigma}\right)^2\right]. \tag{1}$$

The parameter $\sigma$ determines the strength of selection by controlling how rapidly fitness decreases as the phenotype $w$ deviates from the optimal phenotype $w_{opt}$. Increasing $\sigma$ decreases the strength of selection, because it broadens the fitness map around $w_{opt}$ and thus decreases the fitness differences between similar phenotypes. Similar maps are commonly used in evolutionary modeling frameworks, such as Fisher's Geometric model [70, 71] and models of speciation [72], as well as in biophysical models of intermolecular interactions [29], including transcription factor-DNA interactions [46]. However, we emphasize that most of our measures of incongruence depend only on the rank ordering of phenotypic or fitness values, and are therefore independent of the exact shape of the phenotype-fitness map, so long as it is symmetric.

Fig 1B–1F illustrates the application of the phenotype-to-fitness map to a simple three-locus, biallelic genotype-phenotype landscape. This landscape is purely additive, and as a consequence, it exhibits no epistasis and has only one peak (Fig 1B). Applying the

phenotype-to-fitness map (Fig 1C) to this landscape can change the amount and type of epistasis, as well as the location and number of peaks in the resulting fitness landscape, relative to the genotype-phenotype landscape (Fig 1D–1F). It can therefore cause incongruence between genotype-phenotype and fitness landscapes. Whereas this schematic and our analyses below pertain to a single phenotype, extending our model to multiple phenotypes is straightforward, as we later discuss.

## Two-locus biallelic genotype-phenotype landscapes

We first study incongruence using the simplest form of genotype-phenotype landscape that is capable of exhibiting epistasis: a two-locus biallelic landscape. We represent genotypes as binary strings of length $L = 2$ and randomly assign a phenotype $w_i$ to each genotype $i$, which we draw from a uniform distribution between 0 and 1. We then apply the Gaussian phenotype-fitness map (Eq 1) to generate the corresponding fitness landscape. We repeat this process 10,000 times for values of $w_{opt} \in [0, 1]$ (in increments of 0.01), and report the probability that the type of epistasis in the genotype-phenotype landscape is the same as in the fitness landscape. We first differentiate between no magnitude epistasis, positive epistasis, and negative epistasis, and then between no sign epistasis, simple sign epistasis, and reciprocal sign epistasis.

**Incongruence in magnitude epistasis is highest when selection favors low phenotypic values.**   To determine whether the type of magnitude epistasis is the same in the genotype-phenotype landscape and fitness landscape, we calculate the product $\epsilon_{gp} \cdot \epsilon_f$, which will be positive when the two landscapes have the same type of epistasis. We use this product, rather than a discrete categorization, to ensure analytical tractability. We obtain an analytical expression for $\epsilon_{gp} \cdot \epsilon_f$ by assuming $\sigma$ to be large (S1 Appendix, Derivation 1):

$$\epsilon_f \cdot \epsilon_{gp} = \frac{1}{\sigma^2}\left[2\epsilon_m \cdot \epsilon_{gp} + (2w_{opt} - \Sigma_i w_i) \cdot \epsilon_{gp}^2\right], \tag{2}$$

where $w_i$ is the phenotypic value of genotype $i$ with $i \in \{0, 1\}^2$ and $\epsilon_m = w_{00}w_{11} - w_{01}w_{10}$, which is also known as multiplicative epistasis [73].

The probability that the type of epistasis is the same in the genotype-phenotype and fitness landscapes, $P(\epsilon_{gp} \cdot \epsilon_f > 0)$ can be computed using Monte-Carlo methods and yields results in good agreement with the randomly generated genotype-phenotype landscapes when $\sigma$ is large. This is shown in Fig 2, where three trends are immediately apparent. First, for large $\sigma$, the probability that the type of epistasis is the same in the genotype-phenotype landscape and the fitness landscape increases as the optimal phenotype $w_{opt}$ increases, in agreement with the intuition that selection for large phenotypic values leaves the genotype-phenotype landscape mostly unchanged, except for the nonlinear rescaling introduced by the phenotype-fitness map. Second, even when $w_{opt} = 1$, the probability that the type of epistasis is the same in the genotype-phenotype landscape and the fitness landscape is less than one. The reason is the nonlinear rescaling introduced by the phenotype-to-fitness map does not guarantee conservation of the type of magnitude epistasis, even though it does preserve the rank ordering of fitness values. Results obtained with the randomly generated genotype-phenotype landscapes show this effect becomes even more pronounced as $\sigma$ decreases. This means that as the strength of selection for $w_{opt}$ increases, so does the likelihood of landscape incongruence. Finally, as $\sigma$ decreases, the probability of retaining the type of epistasis from the genotype-phenotype landscape in the fitness landscape is not maximized at $w_{opt} = 1$, but rather at a smaller $w_{opt}$ (e.g., at $w_{opt} \approx 0.8$ when $\sigma = 0.01$, Fig 2A). The reason is as $\sigma$ decreases, the fitness function becomes extremely narrow and more phenotypic values are mapped to zero fitness (within

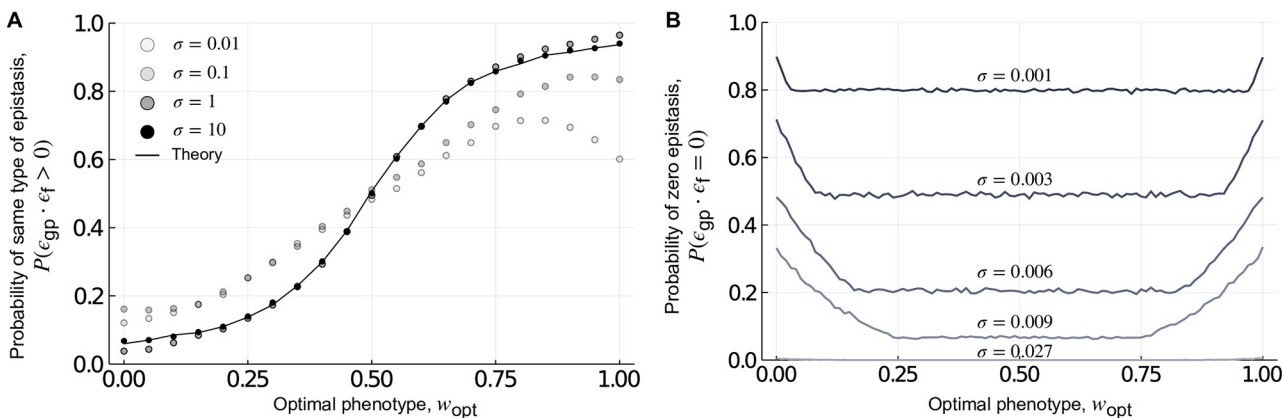

**Fig 2. Local incongruence: Magnitude epistasis.** (A) The probability of retaining the type of epistasis, shown in relation to the optimal phenotype $w_{opt}$. The black line shows the theoretical prediction and the dots show the results from randomly generated genotype-phenotype landscapes for different values of $\sigma$. The theoretical approximation agrees well with the results from randomly generated genotype-phenotype landscapes for large $\sigma$ (i.e., $\sigma \geq 1$). (B) The probability of observing zero magnitude epistasis in the fitness landscape, shown in relation to $w_{opt}$, for different values of $\sigma$, which we selected to show the range of variation in the probability of observing zero epistasis.

computer precision i.e. any fitness $< 5.0 \times 10^{-324} \approx 0$), resulting in cases where the genotype-phenotype landscape exhibits epistasis (i.e., $\epsilon_{gp} \neq 0$), but the fitness landscape does not (i.e., $\epsilon_f = 0$), because all fitness values are zero. Fig 2B shows this is more likely to occur when $\sigma$ is small and as $w_{opt}$ approaches its extreme values of 0 or 1. In these randomly generated landscapes, the probability of obtaining negative epistasis is the same as the probability of obtaining positive epistasis, and the probability of conserving the type of epistasis is independent of the type of epistasis in the genotype-phenotype landscape. In sum, these results show that selection for low or intermediate phenotypic values can modify the genotype-phenotype landscape, such that the resulting fitness landscape exhibits a different type of magnitude epistasis, and this effect is most pronounced when selection is strong and the optimal phenotypic value is low.

**Incongruence in sign epistasis is highest when selection favors intermediate phenotypic values.** Next, we categorized landscapes as exhibiting no sign epistasis, simple sign epistasis, or reciprocal sign epistasis. These three motifs are shown in the center panel of Fig 3, where arrows point from genotypes with a lower phenotypic or fitness value to genotypes with a higher phenotypic or fitness value. Because the presence of sign epistasis only depends upon the partial ordering of the phenotypic values, we expect to retain the motif from the genotype-phenotype landscape in the fitness landscape as $w_{opt} \to 1$. We also expect to retain the motif as $w_{opt} \to 0$, because selecting for $w_{opt} = 0$ simply flips all the arrows from the genotype-phenotype landscape in the fitness landscape, which does not change the categorization of the motif. Thus, we expect the probability of retaining the motif from the genotype-phenotype landscape in the fitness landscape to be "U" shaped and symmetric about $w_{opt} = 0.5$. Fig 3 shows the probability of retaining or changing the motif from the genotype-phenotype landscape in the fitness landscape for 10,000 randomly generated two-locus biallelic landscapes, grouped according to the motif in the genotype-phenotype landscape. These results confirm the expected "U" shape of the probability of retaining the type of epistasis from the genotype-phenotype landscape in the fitness landscape as $w_{opt}$ is varied from 0 to 1.

**Simple sign epistasis cannot be modified into reciprocal sign epistasis.** When the genotype-phenotype landscape has the no sign epistasis motif, selection for an intermediate phenotypic value can transform the landscape into the simple sign epistasis motif or the reciprocal

 

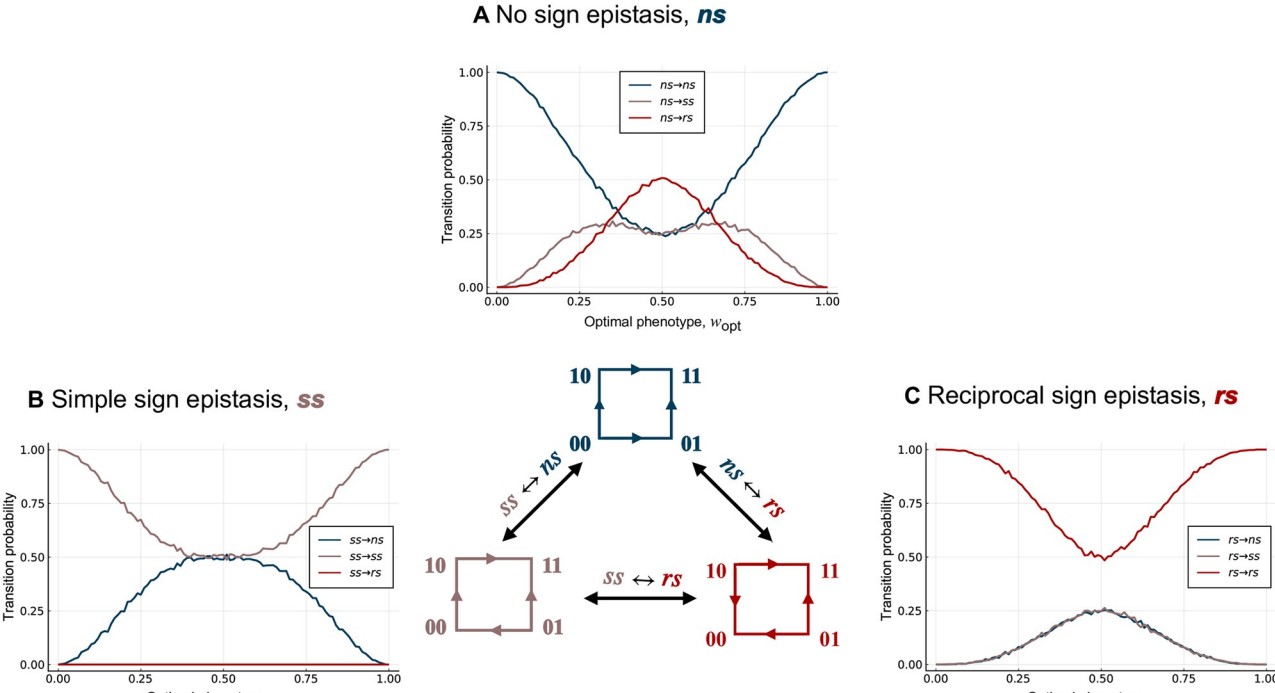

**Fig 3. Local incongruence: Sign epistasis.** The probability of retaining or changing the type of epistasis in the genotype-phenotype landscape, relative to the fitness landscape, shown in relation to the optimal phenotypic value $w_{opt}$. Data are grouped based on whether the genotype-phenotype landscapes exhibits (A) no sign epistasis (blue), (B) simple sign epistasis (brown), or (C) reciprocal sign epistasis (red). The colours of the lines represent the type of epistasis in the resulting fitness landscape. These results are independent of $\sigma$, because they only depend on the rank ordering of fitness values. Notice the "U" shape of the probability of retaining the type of epistasis in each panel.

sign epistasis motif (Fig 3A), in line with recent results on Fisher's Geometric model [74]. When the genotype-phenotype landscape has the simple sign epistasis motif, selection for an intermediate phenotypic value can transform the landscape into a no sign epistasis motif, but not into a reciprocal sign epistasis motif (Fig 3B and S1 Appendix, Proof 1). Because reciprocal sign epistasis is a necessary condition for multiple peaks [5], this implies that genotype-phenotype landscapes with only simple sign epistasis motifs will always give rise to single peaked fitness landscapes, using the phenotype-fitness map considered here. Finally, when the genotype-phenotype landscape has the reciprocal sign epistasis motif, selection for an intermediate phenotypic value transforms the landscape into the no sign epistasis motif or the simple sign epistasis motif with equal probability (Fig 3C and S1 Appendix, Proof 2). Moreover, the probability of retaining the motif from the genotype-phenotype landscape in the fitness landscape is always higher than the probability of changing it when the genotype-phenotype landscape has the simple sign epistasis motif (Fig 3B) or the reciprocal sign epistasis motif (Fig 3C). This is not always true when the genotype-phenotype landscape has the no sign epistasis motif (Fig 3A), because at intermediate $w_{opt}$ the landscape is most likely to transform into the reciprocal sign epistasis motif. Taken together, these results show that selection for intermediate phenotypic values can modify genotype-phenotype landscapes with no sign epistasis into fitness landscapes with sign epistasis and vice versa.

The inferences about pairwise interactions can be carried forward to multi-locus biallelic landscapes because their genotype spaces, which are $L$-dimensional hypercubes, are composed of two-dimensional squares. Due to the adjacency of squares, in the three-locus case, the motifs of four out of the six squares are sufficient to determine the motifs of the rest of the squares,

and for any $L$, the motifs of only $2^{L-2} \cdot (L-1)$ squares are necessary to determine the motifs of all of the remaining squares. This is only a fraction $2/L$ of all the squares in the hypercube (because the total number of faces in an $L$-dimensional hypercube is $2^{L-2} \cdot \binom{L}{2}$), which is clearly minuscule for large $L$. However, pairwise interactions are not sufficient to predict peak patterns, which may result from higher-order interactions [75]. We study these in the next section.

## Multi-locus biallelic genotype-phenotype landscapes

We use the NK model [67] to study multi-locus biallelic genotype-phenotype landscapes (Methods). In this model, each locus in a genotype of length $L$ epistatically interacts with $K$ other loci (whereas $N$ is typically used to denote the number of loci in this model, we use $L$ for consistency with the rest of our text). As corner cases, this model includes Mt. Fuji landscapes [65] when $K = 0$ and House-of-Cards landscapes [66] when $K = L - 1$. For each combination of $L$ and $K$, we use this model to randomly generate a genotype-phenotype landscape. We then apply the Gaussian phenotype-fitness map (Eq 1) to generate a fitness landscape. We repeat this process 10,000 times for $w_{opt} \in [0, 1]$ (in increments of 0.01), and report the average of the absolute change in the number of peaks, i.e., $\langle |p_f - p_{gp}| \rangle$, where $p_f$ is the number of peaks in the fitness landscape and $p_{gp}$ is the number of peaks in the genotype-phenotype landscape. This is our measure of global incongruence. We use the absolute value of the change in number of peaks so that we can average over many realisations of genotype-phenotype landscapes. However, since the sign of change is also important, we discuss that as well in the following sections.

**Mt. Fuji landscapes.** We begin with Mt. Fuji genotype-phenotype landscapes. Because these are single-peaked, selection for a low or intermediate phenotypic value can only maintain or increase the number of peaks from the genotype-phenotype landscape in the fitness landscape. Fig 4A shows this change in the number of peaks, and Fig 4B shows the probability that the number of peaks changes, in relation to $w_{opt}$ for landscapes with $L = 2$ to $L = 8$ loci. These trends are symmetric about $w_{opt} = 0.5$, because Mt. Fuji landscapes are additive, so selecting for $w_{opt} = 0$ is equivalent to selecting for $w_{opt} = 1$ with regard to the change in the number of peaks. The reason is that selecting for $w_{opt} = 0$ flips all of the arrows in the fitness landscape, relative to the genotype-phenotype landscape, which changes the location of the peak, but does not change the number of peaks. This is illustrated in Fig 1B and 1D. For a detailed explanation of the shape of the curves in Fig 4A, see S1 Appendix, Note 1. More obvious is the increase in the number of peaks, and the probability that the number of peaks increases, as $L$ increases, the latter converging to one for all values of $w_{opt}$ except the extreme cases of $w_{opt} = 0$ and $w_{opt} = 1$. Note, however, that a high probability of increase in the number of peaks does not necessarily correspond to a high increase in the number of peaks, as can be seen from the different positions of the maxima in Fig 4A and 4B (S1 Appendix, Note 1). For large $L$ ($>10$), the expected number of peaks in the fitness landscape increases exponentially with $L$ [74]. In sum, these results show that selection for intermediate phenotypic values readily transforms Mt. Fuji genotype-phenotype landscapes, which are smooth and single-peaked, into rugged fitness landscapes, and that this effect is most pronounced for large $L$ and intermediate $w_{opt}$.

**House-of-Cards landscapes.** We next study House-of-Cards genotype-phenotype landscapes. These landscapes are highly rugged, with an average of $\frac{2^L}{L+1}$ peaks [76], whereas the maximum possible number of peaks is $2^{L-1}$. As such, selection for a low or intermediate phenotypic value can either increase or decrease the number of peaks in the fitness landscape, relative to the genotype-phenotype landscape. Fig 4C shows this change in the number of peaks, and Fig 4D shows the probability that the number of peaks changes, in relation to $w_{opt}$ for landscapes with $L = 2$ to $L = 8$ loci. The change in the number of peaks is symmetric about $w_{opt} = 0.5$ for the two-locus case, where the number of peaks does not change as $w_{opt} \to 0$ or

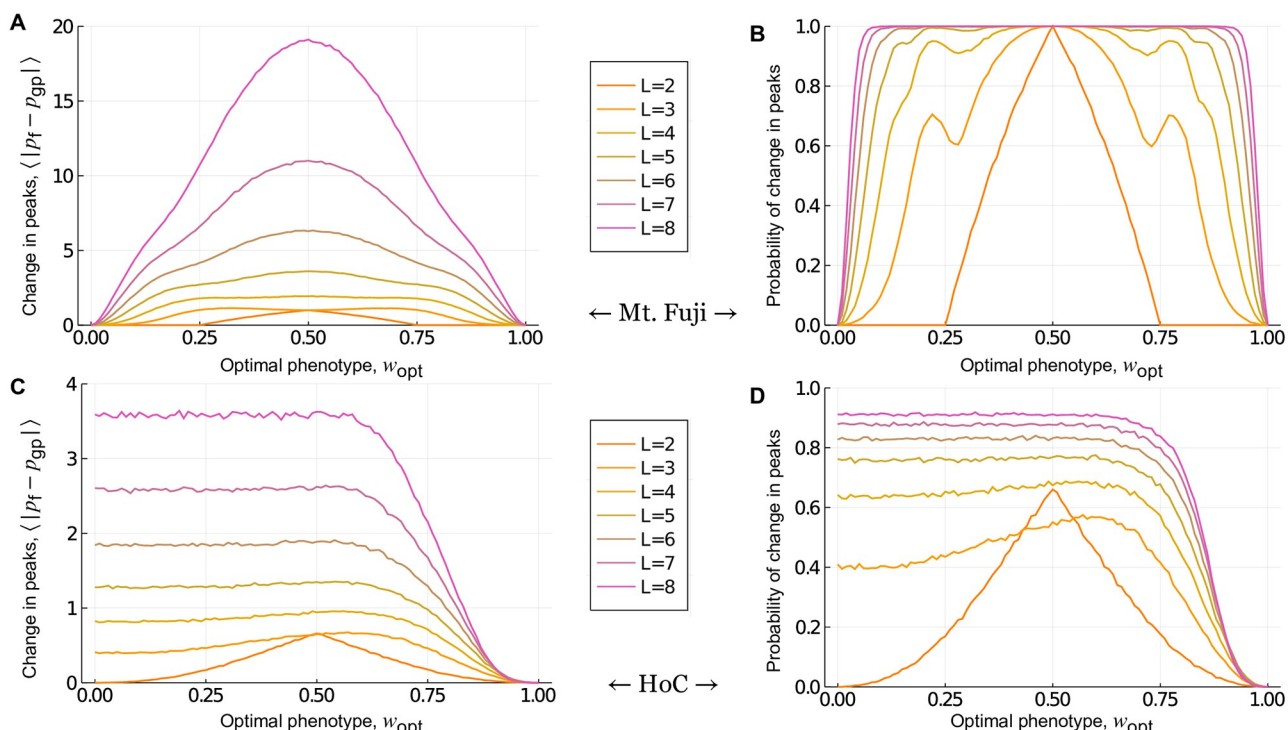

**Fig 4. Global incongruence: Mt. Fuji and House-of-Cards genotype-phenotype landscapes.** The absolute change in the number of peaks and the probability that the number of peaks changes in the fitness landscape, relative to (A,B) Mt. Fuji and (C,D) House-of-Cards genotype-phenotype landscapes, shown in relation to $w_{opt}$ for $L \in \{2, 3..8\}$. These results are independent of $\sigma$, because they only depend on the rank ordering of fitness values.

$w_{opt} \rightarrow 1$. However, this symmetry is lost for $L > 2$. The reason is that although the phenotype-fitness map flips all of the arrows in the fitness landscape when $w_{opt} = 0$, relative to the genotype-phenotype landscape, this does not guarantee conservation of the number of peaks. The number of peaks is jointly determined by the adjacent faces of the hypercube and thus, only very specific changes in the directions of arrows in the genotype-phenotype landscape guarantees conservation of the number of peaks in the fitness landscape (S4 Fig). However, in contrast to Mt. Fuji genotype-phenotype landscapes, the magnitude of change in the number of peaks increases very little with $L$, despite an exponential increase in the maximum number of possible peaks. Moreover, the probability that the number of peaks changes is still less than one for large $L$. Finally, for large $L$, both the change in the number of peaks and the probability that the number of peaks changes are independent of $w_{opt}$, so long as $w_{opt}$ is sufficiently less than one. This observation depends on the probability distribution used to generate these landscapes. Because the $2^L$ phenotypes in the NK model are drawn from a uniform distribution, nearly the same number of these phenotypes will be close to the optimal phenotype $w_{opt}$, so long as $L$ is sufficiently large. Thus, averaging over all possible configurations of the genotype-phenotype landscape yields the same value of $\langle |p_f - p_{gp}| \rangle$ for every $w_{opt}$. For sufficiently large $L$, this value is given by (S1 Appendix, Derivation 2):

$$\langle |p_f - p_{gp}| \rangle \approx \sqrt{\frac{2^L \cdot (L-1)}{2\pi(L+1)^2}} \text{ for } L \gg 1 \tag{3}$$

So far we have focused on the absolute change $\langle |p_f - p_{gp}| \rangle$ in the number of peaks in the fitness landscape, relative to the genotype-phenotype landscape. For House-of-Cards genotype-

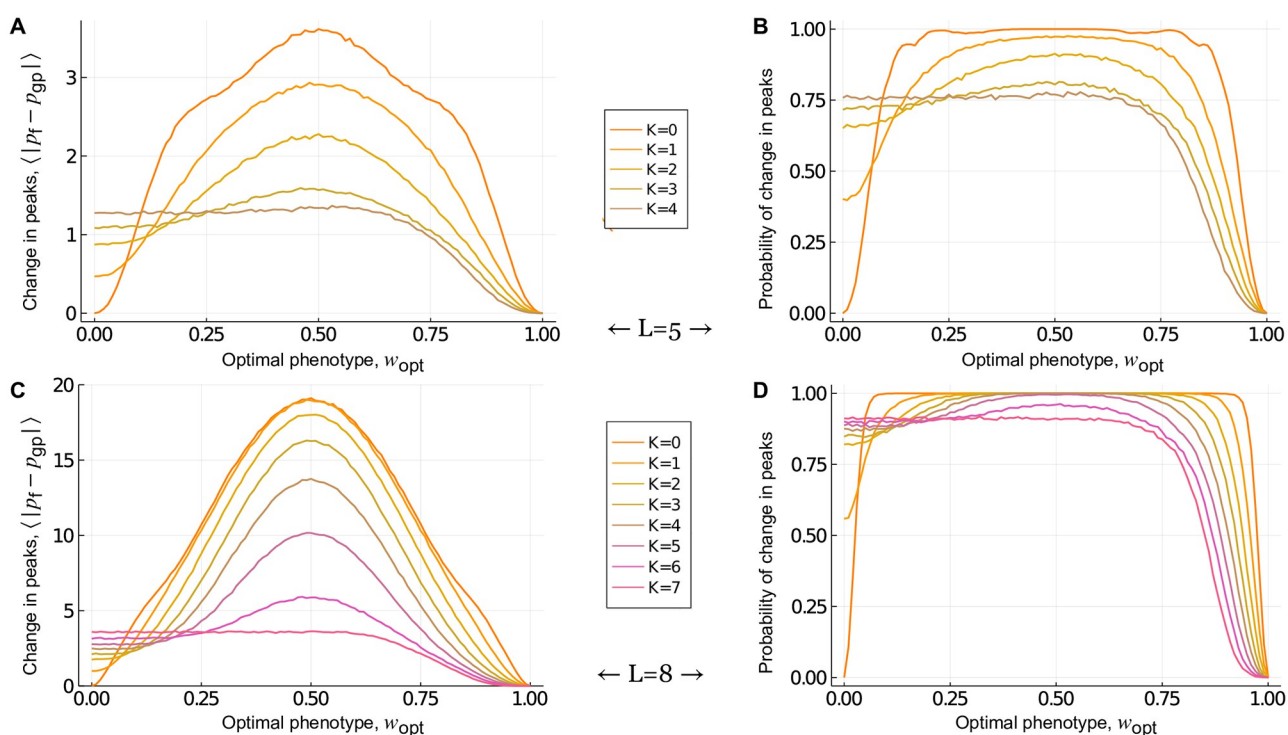

**Fig 5. Global incongruence: NK genotype-phenotype landscapes.** The absolute change in the number of peaks in the fitness landscape, relative to the genotype-phenotype landscape, is shown in relation to $w_{opt}$ for genotypes of length (A) $L = 5$ and (C) $L = 8$, as $K$ increases from zero to $L − 1$. The corresponding probability of change in the number of peaks is shown in relation to $w_{opt}$ for genotypes of length (B) $L = 5$ and (D) $L = 8$, as $K$ increases from zero to $L − 1$. These results are independent of $\sigma$, because they only depend on the rank ordering of fitness values.

phenotype landscapes, selection for a low or intermediate phenotypic value can either increase or decrease the number of peaks. We were therefore interested in finding out which outcome is more likely. While one might expect a decrease to be more likely, due to the extreme ruggedness of House-of-Cards genotype-phenotype landscapes, we find that the number of peaks is equally likely to increase or decrease (S1 Appendix, Proof 3). In sum, these results show that in House-of-Cards genotype-phenotype landscapes, selection for a low or intermediate phenotypic value increases or decreases the number of peaks in the fitness landscape with equal probability, and the severity as well as the probability of this change increases with $L$ and is largely independent of $w_{opt}$.

**Global incongruence decreases as the ruggedness of the genotype-phenotype landscape increases.** Finally, we study NK genotype-phenotype landscapes, which bridge the gap between Mt. Fuji and House-of-Cards landscapes in terms of ruggedness, as $K$ increases from 0 to $L − 1$. Fig 5 shows the absolute change in the number of peaks in the fitness landscape relative to the genotype-phenotype landscape for genotypes of length $L = 5$ and $L = 8$, as $K$ is increased from 0 to $L − 1$. Note the gradual transition from the trends observed for Mt. Fuji genotype-phenotype landscapes (Fig 4A) to those observed for House-of-Cards genotype-phenotype landscapes (Fig 4C) as $K$ increases. From these trends, we conclude four principles of how the ruggedness of an NK genotype-phenotype landscape influences its incongruence with the fitness landscape. As an NK genotype-phenotype landscape becomes more rugged, incongruence (1) loses symmetry about $w_{opt} = 0.5$, (2) becomes less sensitive to $w_{opt}$, and (3) decreases in severity, at least in terms of the absolute change in the number of peaks. Finally, the probability of increasing the number of peaks is always greater than or equal to the

probability of decreasing the number of peaks, with the equality holding for House-of-Cards genotype-phenotype landscapes. This last principle is both intuitive and informative—it tells us that on average, selection for low or intermediate values is more likely to increase the ruggedness of a fitness landscape, relative to the genotype-phenotype landscape. Thus, selection for low or intermediate values is more likely to break than to create phenotypic correlations between mutationally similar genotypes as they map onto fitness, rendering fitness landscapes more rugged than their underlying genotype-phenotype landscapes. In the subsequent sections we address whether and how these results apply to genotype-phenotype landscapes with more than two alleles per locus, specifically in the context of a biophysical model and experimental measurements of transcription factor-DNA interactions.

## Genotype-phenotype landscapes of transcription factor-DNA interactions

Motivated by the common usage of low- and intermediate-affinity transcription factor binding sites in the regulatory portfolios of a diversity of organisms [49, 51, 54–57], we now study the incongruence of genotype-phenotype landscapes of transcription factor-DNA interactions and the corresponding fitness landscapes generated after selection for low or intermediate phenotypic values. In these landscapes, genotypes represent DNA sequences—transcription factor binding sites—and the phenotype of a DNA sequence is the affinity with which it binds a transcription factor [21]. Because the regulatory effects of transcription factor-DNA interactions are partly determined by binding affinity [41, 42] and mutations to transcription factor binding sites can alter binding affinity [48, 77], the topographies of genotype-phenotype landscapes of transcription factor-DNA interactions have important implications for the evolution of gene regulation [21]. We study these landscapes using both a biophysical model and experimental measurements of transcription factor-DNA interactions. We focus on transcription factor binding sites of length $L = 8$, because this is the length of the binding sites assayed by protein binding microarrays [77, 78]—the data used to construct the empirical genotype-phenotype landscapes of transcription factor-DNA interactions [21].

**The mismatch model.**   We first study genotype-phenotype landscapes of transcription factor-DNA interactions generated using the so-called mismatch model [46, 47, 68]. The key assumption of this model is that the binding energy of a DNA sequence is a linear function of the number of mismatches between the sequence (genotype) and a transcription factor's consensus sequence—the sequence it binds with the highest affinity. Further, each mismatch is assumed to have the same energetic cost and these costs combine additively to determine binding energy. This model results in a Mt. Fuji-like, permutation-invariant genotype-phenotype landscape, wherein the phenotype only depends on the number of differences between the genotype and the consensus sequence, but not on which loci in the genotype differ from the consensus sequence. Although this is a simplified model, it provides an opportunity to study the effects of having more than two alleles per locus and serves as a bridge to our analyses of empirical transcription factor-DNA interactions.

To ensure that these results are comparable to the results on the empirical landscapes, we consider the negative of the binding energy as our phenotype, such that sequences that bind more strongly are assigned higher phenotypic values. We assume a phenotypic value of 1 for the genotype that is identical to the consensus sequence. For each mismatch between a genotype and the consensus sequence, we deduct a small positive value $e$, such that the phenotypic value of a genotype with $m$ mismatches is $A_m = 1 - m \cdot e$. Due to the permutation invariance, the genotype-phenotype landscape is highly degenerate, such that the number of genotypes

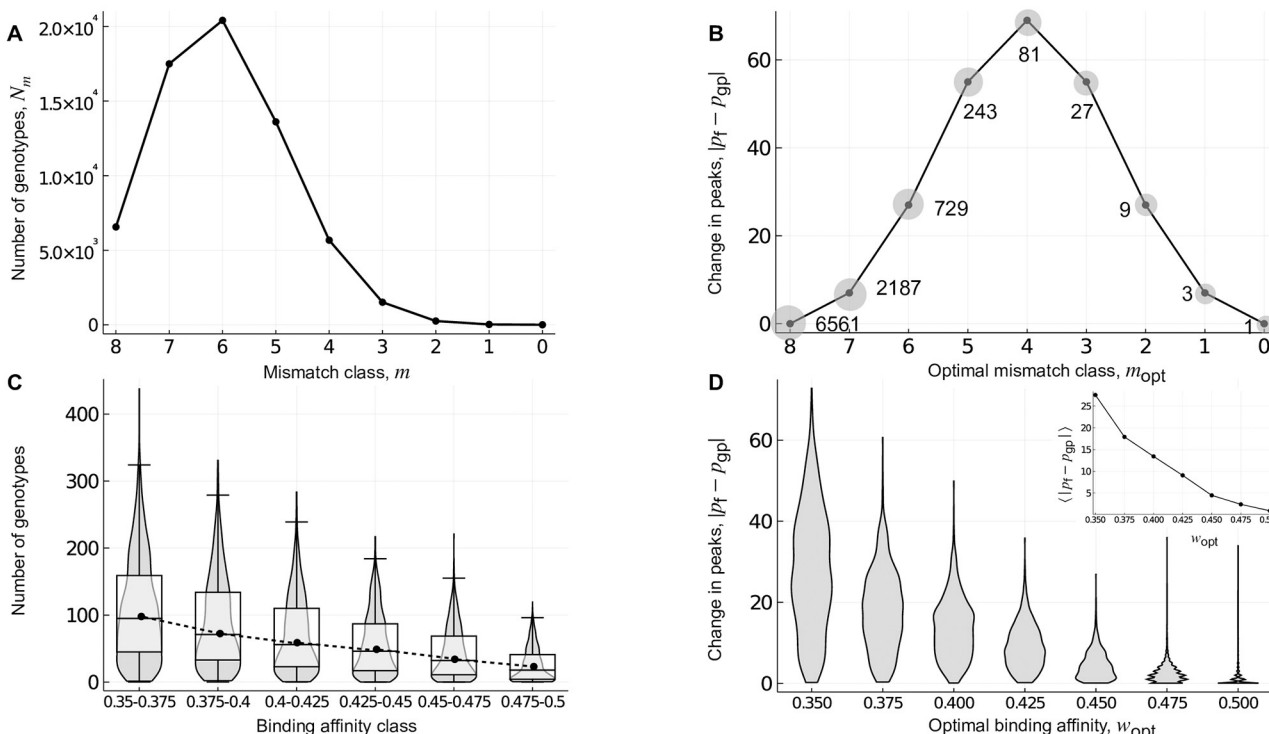

**Fig 6. Global incongruence: Genotype-phenotype landscapes of transcription factor-DNA interactions.** Landscapes constructed using (A,B) the mismatch model and (C,D) experimental measurements from protein binding microarrays for 1,137 eukaryotic transcription factors. (A) The number of genotypes, shown in relation to mismatch class. (B) The absolute change in the number of peaks in the fitness landscape, relative to the genotype-phenotype landscape, shown in relation to the optimal mismatch class $m_{\text{opt}}$. Labels indicate the number of genotypes per peak in the fitness landscape. Note the symmetry in the absolute change in the number of peaks around mismatch class $m_{\text{opt}} = 4$, as well as the tripling of the number of genotypes per peak for each increment in $m_{\text{opt}}$. The grey shaded circles are a schematic representation of the growing width of the peaks. (C) The number of genotypes per binding affinity class, where protein binding microarray $E$-scores are used as a proxy for relative binding affinity. Violin plots show the distribution, and box-and-whisker plots the 25–75% quartiles, across genotype-phenotype landscapes for the 1,137 transcription factors. Closed symbols and the dashed line denote the median of each distribution. (D) Violin plots of the distribution of the absolute change in the number of peaks in the fitness landscape, relative to the genotype-phenotype landscape, shown in relation to the optimal binding affinity $w_{\text{opt}}$ for $\sigma = 0.15$. The inset shows the mean absolute change in the number of peaks, in relation to $w_{\text{opt}}$. The x-axes in (A,B) are arranged such that binding affinity increases when read from left to right, in qualitative agreement with the x-axes in (C,D). The results in panels A-C are independent of $\sigma$.

$N_m$ in a mismatch class $m$ is distributed according to the asymmetric binomial distribution:

$$N_m = (a-1)^m \frac{L!}{(L-m)!m!}, \tag{4}$$

where $a$ is the number of alleles per locus ($a = 4$ for transcription factor binding sites, because they are DNA sequences). Fig 6A shows this distribution for transcription factor binding sites of length $L = 8$. Note that $N_m$ is maximized at $m = 6$.

Fig 6B shows the incongruence between genotype-phenotype landscapes and fitness landscapes of transcription factor-DNA interactions constructed using the mismatch model, under selection for an optimal mismatch class $m_{\text{opt}}$, reported in terms of the absolute change in the number of peaks in the fitness landscape, relative to the genotype-phenotype landscape. Based on our analysis of Mt. Fuji genotype-phenotype landscapes, we anticipated asymmetric incongruence about the mismatch class $m_{\text{opt}} = 6$, because the distribution of the number of genotypes per mismatch class is asymmetric with a maximum at mismatch class $m = 6$ (Fig 6A) and we expected all genotypes in the optimal mismatch class $m_{\text{opt}}$ to be peaks. However, we observe

symmetric incongruence about $m_{opt} = 4$ (Fig 6B). This occurs because $a > 2$, which renders some genotypes in the same mismatch class mutational neighbours. Consequently, the peaks can be broad and include more than one genotype, thus resembling plateaus. Specifically, each genotype has $(a - 1)^m - 1$ mutational neighbours that are in the same mismatch class and are therefore part of the same peak. This leaves $\binom{L}{M}$ clusters of genotypes to be peaks in each mismatch class $m$, an expression that is maximized with $m = 4$ when $L = 8$, thus forcing symmetry in the absolute change in the number of peaks about $m_{opt} = 4$. However, the width of the peaks increases as $(a - 1)^m$, leading to a tripling of peak width for each increment in $m$ (Fig 6B). Thus, even in this idealised genotype-phenotype landscape, features of empirical landscapes of TF-DNA interactions begin to emerge, such as broad peaks.

**Empirical landscapes.** We now study genotype-phenotype landscapes of transcription factor-DNA interactions generated using experimental data from protein binding microarrays [78] (Methods). For all possible DNA sequences of length $L = 8$, these data include an enrichment score ($E$-score) ranging from -0.5 to 0.5 that serves as a proxy for relative binding affinity, such that higher $E$-scores correspond to higher binding affinities [77, 78]. We have previously used these data to construct genotype-phenotype landscapes for 1,137 eukaryotic transcription factors, in which the surface of each landscape was defined by the $E$-score [21]. Due to limitations in the reproducibility of $E$-scores across microarray designs for genotypes that are bound non-specifically or with very low affinity [77, 78], each genotype-phenotype landscape only includes DNA sequences with an $E$-score exceeding a threshold of 0.35, which corresponds to a false discovery rate of 0.001 [77]. As shown in our previous work [21], these landscapes tend to exhibit little, if any, reciprocal sign epistasis and therefore comprise few peaks. As such, they bear resemblance to the genotype-phenotype landscapes constructed using the mismatch model. An important difference, however, is that genotypes in the lower half of the $E$-score range are omitted from each landscape due to the reproducibility issues mentioned above. Fig 6C shows the distributions of the number of genotypes across all 1,137 genotype-phenotype landscapes, grouped into six binding affinity classes. Whereas the lowest binding affinity class contains the most genotypes, we cannot determine if this is the true maximum, because we do not know what these distributions look like for lower binding affinity classes. However, assuming the energetic contribution of each binding site to be additive [68], we expect lower binding affinity classes to have fewer genotypes and the maximum to occur at an intermediate binding affinity class, as can be seen in the mismatch model and other models in literature [43].

Fig 6D shows the incongruence between these 1,137 empirical genotype-phenotype landscapes and their corresponding fitness landscapes, under selection for an optimal binding affinity $w_{opt}$, reported in terms of the absolute change in the number of peaks in the fitness landscape, relative to the genotype-phenotype landscape for $\sigma = 0.15$. Interestingly, the effect of changing $\sigma$ is small (S5B Fig) because increasing $\sigma$ not only decreases the range of variation of fitness values, but also decreases the uncertainty in these values (Methods), leaving the number of peaks largely unchanged. The mean and variance in the absolute change in the number of peaks decreases as $w_{opt}$ increases to 0.5, in line with the intuition that selection for high $w_{opt}$ generates fitness landscapes that are topographically similar to the underlying genotype-phenotype landscape. More precisely, the percentage of landscapes that show a change in the number of peaks decreases from around 99% to 40%, as we go from $w_{opt} = 0.35$ to $w_{opt} = 0.5$. As anticipated from our results with additive genotype-phenotype landscapes (Mt. Fuji and the mismatch model), selection for low or intermediate phenotypic values is more likely to increase than to decrease the number of peaks, although the relative fraction of increase depends upon $w_{opt}$—while around 99% of the landscapes show an increase in peaks for $w_{opt} = 0.35$, this value decreases to around 40% for $w_{opt} = 0.5$. Further, when we separately analyzed

the single-peaked ($\approx 66.75\%$) and multi-peaked ($\approx 33.25\%$) genotype-phenotype landscapes, we found the single-peaked landscapes to be more incongruent (S5A Fig), in line with our results from the previous section. In sum, these results show that selection for low or intermediate phenotypic values tends to increase the ruggedness of a fitness landscape, relative to the underlying genotype-phenotype landscapes, rendering genotype-phenotype landscapes a poor proxy for fitness landscapes under such selection.

**Mismatch model and empirical landscapes show different kinds of global incongruence.**   In contrast to genotype-phenotype landscapes constructed using the mismatch model, the incongruence of genotype-phenotype landscapes constructed using protein binding microarray data is highest for the lowest binding affinity class, rather than an intermediate class. There are three non-mutually exclusive explanations for this. First, these empirical landscapes are not purely additive [21], unlike the landscapes constructed with the mismatch model, so we do not expect perfect symmetry about an intermediate $w_{opt}$. Second, as previously mentioned, protein binding microarray data do not capture the full range of binding affinity, so the lowest binding affinity class in our data ($E$-score = 0.35), which contains the most genotypes (Fig 6C), is unlikely to be the lowest binding affinity class, but rather an intermediate binding affinity class. Third, while the binding affinity of a sequence is highly correlated with the binding affinities of its mutational neighbors [79], this correlation is not perfect, so neighboring genotypes that are in the same mismatch class may not have sufficiently similar binding affinities to be considered part of the same peak, unlike in the mismatch model.

There are two additional differences between the incongruence of the empirical genotype-phenotype landscapes and those constructed using the mismatch model that are worth highlighting. First, in the empirical landscapes, the average height of the peaks is maximised when selecting for intermediate binding affinities and is lowest when selecting for the highest affinity (S6A Fig), whereas peak height is independent of $m_{opt}$ in the mismatch model. Second, in the empirical landscapes, peak width is maximized when selecting for the highest binding affinity ($w_{opt} = 0.5$) (S6B Fig), whereas it is maximized when selecting for the lowest binding affinity ($m_{opt} = 8$) in the mismatch model (Fig 6B). These two differences in incongruence are important for understanding evolutionary simulations on these landscapes, which are the focus of the next section.

## Evolutionary consequences

A key finding of our analyses so far is that selection for low or intermediate phenotypic values is more likely to increase than to decrease the number of peaks in the fitness landscape, relative to the genotype-phenotype landscape. Since the ruggedness of a fitness landscape has important implications for a diversity of evolutionary phenomena [59–62], we now study the evolutionary consequences of this finding. We do so using two metrics: (1) the length $\langle l \rangle$ of a greedy adaptive walk, averaged over all possible genotypes as starting points, and (2) the mean population fitness at equilibrium $\langle f \rangle$ under deterministic mutation-selection dynamics (i.e., assuming an infinite population size). These metrics provide complementary information to measures of landscape ruggedness, such as the number of peaks. To see why, consider our results with the mismatch model, in which the fitness landscape had one or more global peaks of equal height (Fig 6A and 6B). We observed single-peaked fitness landscapes when the optimal mismatch class was $m_{opt} = 0$ or $m_{opt} = 8$, and the most rugged fitness landscapes when the optimal mismatch class was $m_{opt} = 4$. We also observed a tripling of the number of genotypes per peak as $m_{opt}$ increased from 0 to 8, such that the landscape with $m_{opt} = 0$ comprised a single peak with one genotype, the landscape with $m_{opt} = 4$ comprised 70 peaks with 81 genotypes per peak,

and the landscape with $m_{opt} = 8$ comprised 1 peak with 6,561 genotypes. Which landscape topography is most conducive to adaptive evolutionary change?

**NK landscapes.** On NK landscapes, the length of the greedy adaptive walk $\langle l \rangle$ is minimised when $w_{opt} = 0.5$ for $K < L - 1$, whereas for $K = L - 1$ (House-of-Cards landscapes), $\langle l \rangle$ is independent of $w_{opt}$. Moreover, the change in mean fitness at equilibrium is always positive, increases with $L$ and $K$, and tends to be maximized at $w_{opt} = 0.5$ (S1 Appendix, Note 2). In sum, these results show that on biallelic landscapes, selection for an intermediate phenotypic value decreases the length of a greedy adaptive walk and increases mean fitness at equilibrium, despite increasing the overall ruggedness of the fitness landscape, relative to the genotype-phenotype landscape. Whereas previous work has shown that peak accessibility increases with alphabet cardinality ($a$) due to the existence of indirect paths [10, 80], we show below that despite this increased accessibility, results qualitatively similar to the biallelic case ($a = 2$) also hold for the multi-allelic case of the mismatch model and of the empirical landscapes of TF-DNA interactions ($a = 4$).

**Mismatch model.** In the fitness landscapes generated using the mismatch model, the length of the greedy adaptive walk averaged over all starting genotypes is given by

$$\langle l \rangle = \frac{\sum_{m=0}^{L}(a - 1)^m \cdot \frac{L!}{(L - m)!m!} \cdot |m_{opt} - m|}{a^L}. \tag{5}$$

Fig 7A shows this expression for $L = 8$ and $a = 4$. Due to the additive nature and degeneracy of this genotype-phenotype landscape, the fitness landscapes have monotonically decreasing

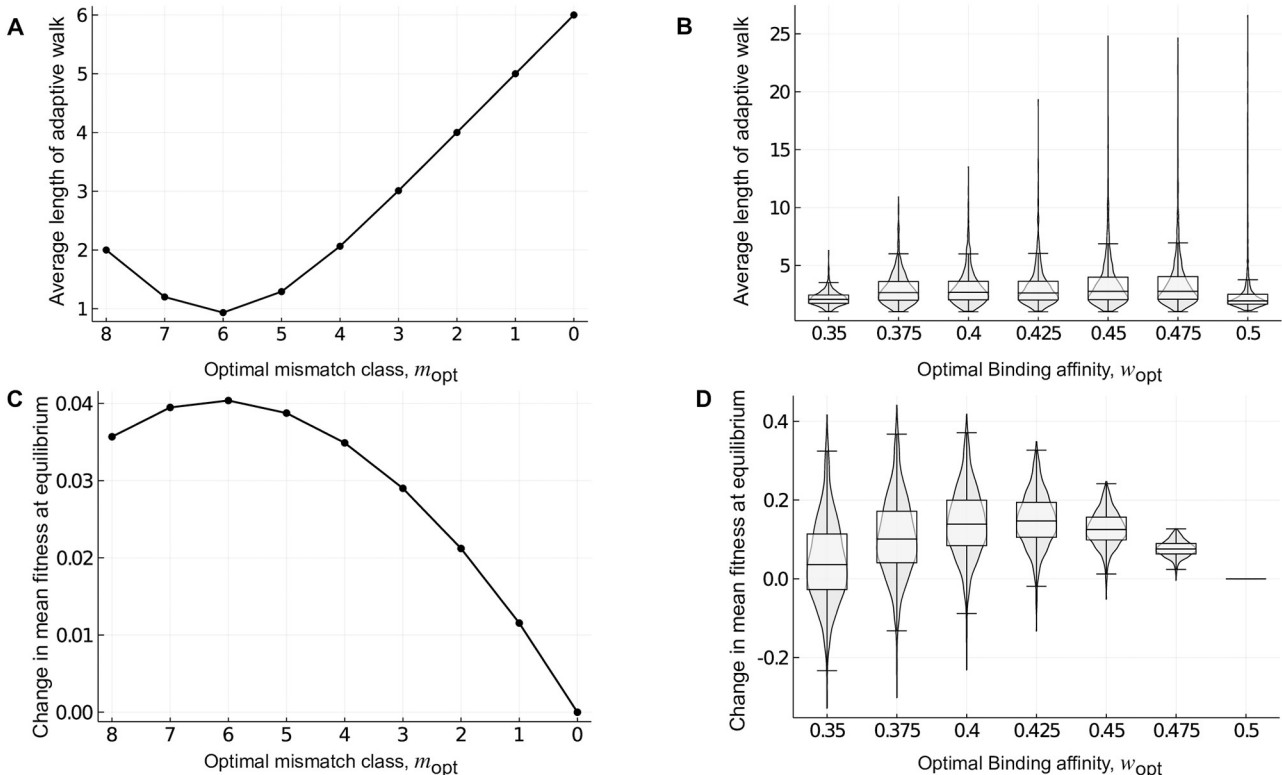

**Fig 7. Rugged fitness landscapes need not impede adaptation.** The average length of an adaptive walk $\langle l \rangle$ and the change in mean population fitness at equilibrium $\langle f \rangle$ is shown for landscapes of transcription factor-DNA interactions generated using (A,C) the mismatch model for $e = 0.05$ and (B,D) protein binding microarray data. In (B,D), violin plots show the distribution, and box-and-whisker plots the 25–75% quartiles, across the 1,137 empirical landscapes for each optimal binding affinity $w_{opt}$. The large variability of $\langle l \rangle$ at intermediate and high $w_{opt}$ is a consequence of the random diffusion of the population on non-peak plateaus, which results in longer walks. In (C), $\mu = 0.1$, $\sigma = 1$ while in (D) $\mu = 0.1$, $\sigma = 0.15$.

fitness values as the mutational distance from any peak increases. Therefore, the length of any individual greedy adaptive walk is simply the absolute difference of the mismatch class of the starting genotype and that of the optimal mismatch class (i.e., $|m_{opt} - m|$). As such, the average length of the greedy adaptive walk is minimized when selecting for $m_{opt} = 6$ (Fig 7A), because this maximizes the number of genotypes in adaptive peaks (Fig 6B; 28 global peaks × 729 genotypes per peak). If instead, we chose to start the greedy walk from only non-peak genotypes, the walk would still be minimised for $m_{opt} = 6$, because this class has the maximum number of genotypes that are Hamming distance one away from the peak genotypes. Recall that in each fitness landscape generated upon selection for $m_{opt}$, all the peaks are of the same height and therefore, the greedy walk always terminates on a global peak. This is in contrast with the NK landscapes, and as we will see below, the empirical landscapes.

For each $m_{opt}$, we next calculated the change in mean fitness at equilibrium under deterministic mutation-selection dynamics, relative to the fitness landscape generated for $m_{opt} = 0$. To do so, we exploit the permutation-invariance of these landscapes to group genotypes into a lower-dimensional state space defined by mismatch class (Methods). Specifically, we construct a transition matrix that defines the probability that a genotype from one mismatch class mutates into another, based on the frequency and fitness of genotypes in each mismatch class. We iterate this matrix until we reach steady state, which is guaranteed by the Frobenius-Perron theorem to be independent of initial conditions, because the matrix is irreducible [81]. Fig 7C shows the change in mean fitness at equilibrium in relation to the optimal mismatch class $m_{opt}$ = 0. Our first observation is that the change in mean fitness is always positive for $m_{opt} > 0$ when $\mu = 0.1$ and $\sigma = 1$, even though selection for such mismatch classes always causes an increase in the number of peaks in the fitness landscape, relative to the underlying genotype-phenotype landscape (Fig 6B). Our second observation is that the change in mean fitness relative to $m_{opt}$ is always unimodal. For example, selection for $m_{opt} = 6$ maximizes the change in mean fitness when $\mu = 0.1$ and $\sigma = 1$ (Fig 7C). More generally, the $m_{opt}$ that maximizes the change in mean fitness depends on the interplay between the mutation rate $\mu$ and the strength of selection $\sigma$. See S10 Fig for the phase diagram. When selection is strong or mutation is weak, mismatch class $m_{opt} = 8$ maximizes the change in mean fitness. As the strength of selection decreases or the mutation rate increases, the mismatch class that maximizes the change in mean fitness decreases from $m_{opt} = 8$ to $m_{opt} = 7$ and then to $m_{opt} = 6$, where it remains for weak selection or high mutation rates. This is because when $\sigma$ is small (i.e., selection is strong), it is costly to step down from a peak and therefore, selecting for the class with the broadest peak ($m_{opt} = 8$) leads to the highest equilibrium mean population fitness. As $\sigma$ increases and selection becomes weaker, it is no longer as costly to step down from a peak and therefore, selecting for the class with the maximum number of genotypes in peaks ($m_{opt} = 6$) maximizes mean fitness at equilibrium. While this effect is reminiscent of the "survival of the flattest" [82] phenomenon, the crucial difference is that in the mismatch model, the peaks always have the same height and thus, there is no trade-off between peak height and width. Another way of altering the strength of selection is by changing $e$, the energetic cost of a mismatch. Larger $e$ corresponds to stronger selection and the phase diagram changes accordingly (S11 Fig).

**Empirical landscapes.** The empirical genotype-phenotype landscapes are topologically more complex [79, 83] than the genotype-phenotype landscapes generated with the mismatch model, which are regular graphs (i.e., every genotype has $(a - 1) \cdot L$ mutational neighbors). We therefore used simulations to calculate the average length of a greedy adaptive walk $\langle l \rangle$ in these empirical landscapes [84], initiating the walks from all non-peak genotypes in the fitness landscape. Moreover, each binding affinity measurement ($E$-score) in the empirical landscapes is associated with a noise threshold that is used to determine whether two genotypes truly differ in phenotype (Methods). This noise threshold can cause landscapes to have large non-peak

plateaus, in which many genotypes have indistinguishable fitness. We therefore modified the greedy adaptive walk such that when a non-peak plateau was encountered, we chose a random mutational neighbor of indistinguishable fitness for the next step in the walk, disallowing reversion mutations. We repeated this process until the plateau was traversed and a sequence with higher fitness was reached. Finally, we terminated the walk when a peak sequence was reached. Therefore, the walk was primarily a deterministic greedy walk, with some stochasticity due to the non-peak plateaus.

Fig 7B shows $\langle l \rangle$, averaged over 100 simulations of the adaptive walk from each initial condition, in relation to $w_{opt}$. In contrast to the mismatch model, $\langle l \rangle$ is shortest for $w_{opt} = 0.35$ and $w_{opt} = 0.5$ and slightly higher for intermediate $w_{opt}$. This is because, as in the mismatch model, $\langle l \rangle$ is correlated with the total number of genotypes in peaks, which depends upon both the number of peaks and their widths. Whereas selecting for $w_{opt} = 0.35$ leads to the largest number of peaks (Fig 6D), selecting for $w_{opt} = 0.5$ leads to the broadest peaks (S6B Fig), thus explaining the minimisation of $\langle l \rangle$ when selecting for these extreme phenotypes. We note that in the absence of plateaus (i.e., when the noise threshold is zero), $\langle l \rangle$ increases monotonically with $w_{opt}$, because in this case, $\langle l \rangle$ is inversely correlated with the number of peaks in the fitness landscape, which decreases monotonically with $w_{opt}$. Regardless of the noise threshold, when selecting for low or intermediate phenotypic values, most walks terminate at a local, rather than the global fitness peak. However, these local peaks tend to be nearly as high as the global peak, especially when selecting for intermediate phenotypic values (S9 Fig). These results hold for all noise thresholds, and are therefore not a consequence of the existence of plateaus in the genotype-phenotype landscapes (see S1 Appendix, Note 2 for explanation).

Next, we simulated deterministic mutation selection dynamics (Methods). For each $w_{opt}$, Fig 7D shows the change in mean fitness at equilibrium for $\mu = 0.1$, $\sigma = 0.15$, relative to the fitness landscape generated after selecting for $w_{opt} = 0.5$. As in the mismatch model, the change in mean fitness tends to be positive, despite the increase in the number of peaks caused by selection for low or intermediate phenotypic values (Fig 6D). Moreover, the mean change in fitness at equilibrium is maximized when selection favors an intermediate phenotypic value ($w_{opt} = 0.425$ for $\mu = 0.1$, $\sigma = 0.15$). This can be explained by the average peak heights and widths that occur when selecting for intermediate phenotypic values (S6A and S6B Fig for $\sigma = 0.15$). Exactly which $w_{opt}$ maximizes the change in mean fitness depends on the interplay between the mutation rate $\mu$ and the strength of selection $\sigma$ (S12 Fig), converging on 0.425 for large $\mu$ and $\sigma$, similar to the convergence seen at $m_{opt} = 6$ for the mismatch model.

## Discussion

Non-linear relationships between phenotype and fitness have been observed in a diversity of biological systems [28, 31, 33, 35, 85, 86], often reflecting a trade-off between the costs and benefits of a particular phenotype, such as antibiotic resistance [87, 88]. It is well established that these non-linearities, either in the genotype-phenotype or phenotype-fitness map are a cause of epistasis [36–38, 74]. Moreover, when mutations have epistatic interactions in their contribution to phenotype, a non-linear phenotype-fitness map can change the form of these interactions from negative to positive, or vice versa [16], as well as introduce or remove sign epistasis [85]. Our work complements these empirical observations and expands upon previous theoretical work [74, 89], by systematically quantifying how and how often selection for a low or intermediate phenotypic value introduces or removes epistasis in the fitness landscape. Specifically, we show that the probability of changing the type of magnitude epistasis (e.g., positive to negative) is highest when selecting for low phenotypic values and the probability of introducing or removing sign epistasis is highest when selecting for intermediate phenotypic

values. Further, we show that the simple sign epistasis motif cannot be converted into reciprocal sign epistasis, implying that genotype-phenotype landscapes with only simple sign epistasis motifs will remain single peaked and globally congruent to their corresponding fitness landscapes.

Another key finding of our analysis is that selection for low or intermediate phenotypic values is more likely to increase than to decrease the number of peaks, with the probability of the two types of change being equal only in House-of-Cards genotype-phenotype landscapes. This means that additive genotype-phenotype landscapes will tend to be incongruent with their fitness landscapes, whereas rugged genotype-phenotype landscapes will not. While increased landscape ruggedness is typically thought to frustrate the evolutionary process, because it limits the amount of adaptive phenotypic variation mutation can bring forth [4], our evolutionary simulations show this need not be the case. Specifically, we find that the rugged fitness landscapes caused by selection for low or intermediate phenotypic values comprise local adaptive peaks that are nearly as tall as the global adaptive peak. Moreover, these local peaks tend to be accessible from throughout the landscape via a small number of sequential mutations that monotonically increase fitness. As a result, the mean population fitness at equilibrium is almost always higher when selecting for low or intermediate phenotypic values than when selecting for a high phenotypic value.

Finally, while there have been several attempts at investigating genotype-phenotype-fitness landscapes in the past [74, 90, 91]—some models have been very specific to the system of interest and others are agnostic to any mechanistic details [92]. We tried to bridge this gap, by applying a Fisher's Geometric model-like phenotype-fitness function to biophysically motivated and empirically determined genotype-phenotype landscapes. Further, our results on the mismatch model and the 1,137 landscapes of TF-DNA interactions may help to explain the prevalence of low- and intermediate-affinity binding sites in the control of gene expression. Prior work has suggested an entropic argument [93]: As with certain RNA secondary structures [94] or regulatory circuit motifs [95], low- and intermediate-affinity binding sites appear more frequently simply because they are more findable. That is, because a transcription factor binds more distinct DNA sequences with low or intermediate affinity than with high affinity, low- and intermediate-affinity binding sites are more likely to evolve to control gene expression. Our work complements this arrival of the frequent argument [96] by showing that low- and intermediate-affinity binding sites are not only more likely to arise *de novo* due to their increased frequency, selection for such sites also generates fitness landscapes that are more conducive to adaptation—in terms of increased mean fitness at equilibrium and decreased average length of adaptive walks, than fitness landscapes that were generated by selection for high affinity binding sites.

We made several modeling assumptions to simplify our analyses of transcription-factor DNA interactions, the relaxation of which may open new avenues for future research. First, we assumed a single nonlinearity in the relationship between genotype, phenotype, and fitness. As noted by Domingo et al. [37], from transcription to RNA processing, translation, and protein folding and all the way up to protein activity and cellular fitness, there are many layers of biological organization where the effects of a mutation can be transformed. To date, our knowledge of how such nonlinearities combine to modify genotype-phenotype landscapes is based on a small number of experimental studies (e.g., ref [16]). A systematic theoretical analysis could build off the work presented here, for example by incorporating the sigmoidal relationship between binding site occupancy and binding affinity in modeling transcription factor-DNA interactions [97], such that fitness depends nonlinearly on occupancy, rather than affinity. Other nonlinearities, such as those caused by transcription factor cooperativity [97], could also be included.

Second, we assumed that selection acts directly on a single phenotype—binding affinity. While this assumption is common in models of the evolution of transcription factor binding sites [21, 43, 46] and is supported by empirical data [43, 44], the relationship between binding affinity and fitness is not so direct, because it is modulated by gene expression. Gene expression depends on a variety of factors, including the presence, arrangement, and affinities of binding sites for other competing or cooperating transcription factors in promoters and enhancers [98], as well as local sequence context [99], chromatin context [100], DNA methylation [101], and local transcription factor concentrations [102]. Existing modeling frameworks that relate the architecture of entire regulatory regions to gene expression patterns may provide a path forward [103], facilitating the study of landscape incongruence when fitness depends upon the multitude of molecular phenotypes characteristic of eukaryotic gene regulation. Alternatively, our modeling framework could be extended to include multiple phenotypes by defining fitness in terms of the differences between a vector of phenotypes and a vector of optimal phenotypes, rather than the scalars considered here. Incongruence could then be quantified between the fitness landscape generated by selecting for the highest phenotypic value of each phenotype and that generated by selecting for a combination of low and intermediate values of the phenotypes. Our results correspond to a special case of this extended model, wherein all phenotypes except one are exactly attuned to their optimal values.

Third, we assumed a Gaussian phenotype-fitness map, which is commonly employed in a diversity of modeling frameworks [29, 70, 72, 104], including those for transcription factor-DNA interactions [46]. Alternative symmetric phenotype-fitness maps (e.g., ref. [105]) will only affect our results quantitatively, because many of our findings, such as the changes in sign epistasis motifs and number of peaks, only depend on the rank ordering of fitness values. However, we expect asymmetric phenotype-fitness maps, such as those uncovered in experimental studies of biological systems such as viruses [30] and yeast [106], to affect our results qualitatively. For example, in our analyses of NK landscapes, we often observed symmetries in incongruence around an intermediate phenotypic value. These symmetries will almost certainly be lost. Understanding how asymmetric phenotype-fitness maps affect the incongruence of genotype-phenotype landscape is therefore an outstanding challenge.

Finally, our study may open new lines of research on dynamic genotype-phenotype and fitness landscapes [107–113]. For example, whereas we studied selection for a fixed phenotypic optimum, this optimum may in fact change in space or in time. Our results imply that even gradual changes in the phenotypic optimum may lead to abrupt changes in fitness landscape topography, which may have implications for an evolving population's ability to track this optimum and thus for population persistence and extinction. Moreover, because our measures of incongruence can be applied to any pair of landscapes so long as they are defined over the same set of genotypes, they are also applicable whenever a phenotype is mapped non-linearly to another phenotype. Ideally, we would be able to make inferences about phenotypic architecture based on the topographical properties of higher-level phenotypic or fitness landscapes—as was previously done for an antibiotic resistance phenotype [114]. However, because the phenotype-fitness map we study is not invertible, we can only make such inferences in limited cases. For instance, when the fitness landscape is single-peaked, we can make the probabilistic inference that the underlying genotype-phenotype landscape is also likely to be single-peaked, because the phenotype-fitness map is more likely to increase than to decrease the number of peaks. In contrast, when the fitness landscape has multiple peaks, we can only infer that the underlying genotype-phenotype landscape does not solely comprise simple sign epistasis motifs. Beyond that, we cannot infer the topographical properties of the underlying genotype-phenotype landscape. It could be smooth or rugged.

Additionally, our measures may shed light on the kinds of topographical alterations induced by fluctuating environmental factors, such as DNA methylation [101] or the presence of protein partners [115] on transcription factor-DNA interactions. As our ability to experimentally interrogate such complexities in the relationship between genotype, phenotype, and fitness continues to improve, we anticipate a sharpened focus on landscape dynamics and their implications for the evolutionary process.

## Methods

### Epistasis

In a genotype-phenotype landscape, we classify the type of magnitude epistasis between a pair of loci using the following linear combination of phenotypic values:

$$\epsilon_{gp} = w_{00} + w_{11} - w_{01} - w_{10}, \tag{6}$$

where $w_i$ represents the phenotype of genotype $i \in \{0, 1\}^2$. When $\epsilon_{gp} = 0$, there is no magnitude epistasis, because the phenotypic effects of the two alleles combine additively; when $\epsilon_{gp} > 0$, there is positive epistasis, because the phenotypic effects of the two alleles are greater than expected based on their individual phenotypic effects; when $\epsilon_{gp} < 0$, there is negative epistasis, because the phenotypic effects of the two alleles are less than expected based on their individual phenotypic effects.

Analogously, in the fitness landscape, we classify the type of magnitude epistasis between a pair of loci using the following linear combination of fitness values:

$$\epsilon_{f} = F_{00} + F_{11} - F_{01} - F_{10}, \tag{7}$$

where $F_i$ represents the fitness of the corresponding genotype $i$. As in the genotype-phenotype landscape, $\epsilon_{f} = 0$, $\epsilon_{f} > 0$, and $\epsilon_{f} < 0$ indicate additive, positive, and negative epistatic interactions among loci, respectively.

We use $\epsilon_{gp}$ and $\epsilon_{f}$ to calculate the fraction of pairs of loci that have the same type of epistasis in the genotype-phenotype landscape and the fitness landscape, which we determine as the product of $\epsilon_{gp}$ and $\epsilon_{f}$. This is because the type of epistasis is the same in the two landscapes when $\epsilon_{gp} \cdot \epsilon_{f} > 0$. While it is theoretically possible for both $\epsilon_{gp}$ and $\epsilon_{f}$ to be 0, in which case the type of epistasis would be the same in the two landscapes yet the condition $\epsilon_{gp} \cdot \epsilon_{f} > 0$ would not be satisfied, this never happens in practice.

### NK Landscapes

We constructed the NK landscapes using the adjacency neighbourhood scheme, wherein each locus $i$ of a genotype of length $L$, interacts with $K$ adjacent loci to the right of locus $i$ and $0 \leq K \leq L - 1$. We used periodic boundary conditions, such that the $L$-th locus interacts with the first $K$ loci, and so on.

The phenotype $w(\tau)$ of genotype $\tau$ is computed as the sum of the individual contributions of all loci, each of which depends on $K$ other interacting loci:

$$w(\tau) = \Sigma_{i=1}^{L} f(\tau_i; \tau_i^1, \tau_i^2 \ldots \tau_i^K), \tag{8}$$

where $f(\tau_i; \tau_i^1, \tau_i^2 \ldots \tau_i^K)$ represents the contribution of the $i$-th locus, which depends on $K$ other loci $\tau_i^1, \tau_i^2 \ldots \tau_i^K$. We drew the contributions of each of the $2^{K+1}$ possible configurations from a uniform distribution between 0 and 1. Finally, we re-scaled the phenotypic values by subtracting the minimum and dividing by the maximum, such that they fell between 0 and 1.

## Empirical genotype-phenotype landscapes of transcription factor-DNA interactions

We studied the incongruence of 1,137 genotype-phenotype landscapes of transcription factor-DNA interactions. The procedure for constructing these landscapes has been described elsewhere [21]. In brief, each landscape corresponds to a single transcription factor, the genotypes it contains represent DNA sequences of length $L = 8$ that specifically bind the transcription factor, and the phenotype of each sequence is a quantitative proxy for relative binding affinity, which defines the surface of the landscape. These phenotypes are reported as enrichment scores (*E*-scores) derived from protein binding microarrays. In each landscape, two genotypes are considered mutational neighbors if they differ by a single small mutation, specifically a point mutation or a small indel that shifts an entire contiguous binding site by a single base [79]. We performed all analyses on the dominant genotype network (i.e., the largest connected component of the network), which comprises the vast majority of genotypes in each landscape [21].

We used the Genonets Python package (version 0.31) to characterize the topographical properties of the empirical genotype-phenotype landscapes [116]. Specifically, we used this package to compute the number of peaks per landscape and the number of genotypes in the peaks of each landscape. These calculations rely on a noise threshold $\delta$, which is used to determine whether two genotypes actually differ in phenotype. For each transcription factor, we used the value of $\delta$ reported in ref. [21], which was derived from a comparison of binding affinity measurements across two protein binding microarray designs.

While characterizing the topographies of the resulting fitness landscapes, we also had to transform $\delta$ following the rules of error propagation. Accordingly, the noise in fitness values, $dF$, depends on the noise in the phenotypic values $dw = \delta$ as follows:

$$dF = -2F \cdot \frac{w - w_{\text{opt}}}{\sigma^2} \cdot \delta. \tag{9}$$

To compute the number of peaks and their widths in the fitness landscapes, we adapted Genonets to specify different noise threshold values ($dF$) for each genotype.

## Mutation-selection dynamics for the mismatch model

We grouped genotypes according to their mismatch class $m$ and iterated a series of selection and mutation steps until the population reached an equilibrium distribution. In each selection step, the frequency of each mismatch class $X_m$ was scaled by its fitness $F_m$ and then normalized:

$$X_m^S(t+1) = \frac{F_m \cdot X_m(t)}{\Sigma_i F_i \cdot X_i}, \tag{10}$$

where $X_m^S(t+1)$ is the frequency of the mismatch class $m$ after the selection step.

The selection step was followed by a mutation step, in which genotypes could either mutate within their mismatch class or mutate to an adjacent mismatch class. The total mutation probability is $\mu$, so with probability $1 - \mu$, the genotype does not mutate. The frequency of each mismatch class was thus updated as

$$X_m^M(t+1) = \left[ \mu \cdot \frac{L - (m-1)}{L} \right] \cdot X_{m-1}^S(t+1) + \\ \left[ (1-\mu) + \mu \cdot \frac{(a-2) \cdot m}{(a-1) \cdot L} \right] \cdot X_m^S(t+1) + \left[ \mu \cdot \frac{m+1}{(a-1) \cdot L} \right] \cdot X_{m+1}^S(t+1), \tag{11}$$

where $X_m^M(t + 1)$ is the frequency of the mismatch class $m$ after the mutation step and $\mu$ is the mutation probability. The mutation step implicitly accounts for the different number of genotypes per mismatch class. Finally, $X_m(t + 1)$ was set to $X_m^M(t + 1)$ and $t$ was incremented. This process was repeated until equilibrium.

### Mutation-selection dynamics for the empirical landscapes

The empirical landscapes comprise far fewer genotypes than the mismatch landscape. We therefore defined the state space of the empirical landscapes in terms of the individual genotypes in each landscape, merging each genotype with its reverse complement [79]. The mutational neighbors of each genotype were those that differed by a single small mutation, namely a point mutation or an indel that shifted an entire contiguous binding site by a single base [79]. In this analysis, we did not account for uncertainties in the fitness values, because these do not influence our results in this evolutionary regime (see below).

The recursion relation for mutation-selection dynamics in discrete time is

$$x_i(t + 1) = \sum_j \mu^{d(\tau_i,\tau_j)} \cdot (1 - \mu)^{L - d(\tau_i,\tau_j)} \frac{f_j}{\bar{f}(\vec{x}, t)} x_j(t), \tag{12}$$

where $\vec{x}$ is the vector of genotype frequencies, $x_i$ is the frequency of the $i$th genotype, $\mu$ is the probability of a single point mutation or a small indel mutation, $L$ is the length of the genotypes, $d(\tau_i, \tau_j)$ is the minimum of the mutational distance between genotypes $\tau_i$ and $\tau_j$ and between genotypes $\tau_i$ and the reverse compliment of $\tau_j$, $f_i$ is the fitness of the $i$th genotype, and $\bar{f}(\vec{x}, t)$ is the mean population fitness at time $t$.

We linearized the dynamics by substituting $z_i(t) = \frac{x_i(t)}{\prod_{\tau=1}^{t-1} \bar{f}(\vec{x}, \tau)}$, yielding

$$z_i(t + 1) = \sum_j \mu^{d(\tau_i,\tau_j)} (1 - \mu)^{L - d(\tau_i,\tau_j)} f_j z_j(t), \tag{13}$$

from which we retrieved the normalized genotype frequencies with $\vec{x} = \vec{z}/(\sum_i z_i)$. In matrix form, the dynamics are

$$\vec{z}(t + 1) = M \cdot S \cdot \vec{z}(t), \tag{14}$$

where the mutation matrix $M$ has elements $M_{ij} = \mu^{d(\tau_i,\tau_j)} \cdot (1 - \mu)^{L - d(\tau_i,\tau_j)}$ and the selection matrix $S$ is a diagonal matrix with $S_{ii} = f_i$, where $f_i$ is the fitness of sequence $\tau_i$. We calculated the eigenvector corresponding to the largest eigenvalue of $M \cdot S$ to determine the equilibrium state of the dynamics, which is guaranteed by the Frobenius-Perron theorem to be unique and stable.

Because the fitness values have some uncertainty around them, we additionally performed a sensitivity analysis by adding Gaussian noise to the fitness values, with standard deviation equal to $1/3 \cdot dF$, such that 99.7% of the sampled fitness values fell within $F \pm \delta$. Our results are robust to these perturbations, with the exception of two parameter combinations—$\sigma = 0.1$, $\mu = 0.001$ and $\sigma = 0.1$, $\mu = 0.0025$. For small $\sigma$, $dF$ is large and therefore such sensitivity is expected. However, the quantitative changes in our results were small, leading us to conclude that fitness uncertainties do not significantly influence our results in this evolutionary regime.

## Supporting information

**S1 Appendix. Derivations, proofs and notes.**
(PDF)

**S1 Fig. Simple sign epistasis motif cannot be changed into a reciprocal sign epistasis motif.** A simple sign epistasis motif (shown in brown) cannot be changed into a reciprocal sign epistasis motif, for any $w_{opt}$. The no sign epistasis motif is shown in blue. The remaining colours represent the neighbourhood of the phenotypic values ($w_i$) corresponding to each genotype ($g_i$), where $i \in \{1, 2, 3, 4\}$ and the genotypes are labelled in descending order, such that the genotype with the highest phenotypic value is labelled 1, and so on. The four bottom arrows point to the transformed motifs when $w_{opt}$ belongs to the neighbourhood from which the arrow emanates.
(PDF)

**S2 Fig. Reciprocal sign epistasis motif can be changed into the no sign epistasis motif and the simple sign epistasis motif with equal probability.** Selection for $w_{opt}$ transforms the reciprocal sign epistasis motif (shown in red) into the no sign epistasis motif (shown in blue) and the simple sign epistasis motif (shown in brown) with equal probability. The neighbourhood of each phenotypic value is shown in a different colour. For the no sign epistasis motif to emerge (top), the fitness values need to be "separated", while for the simple sign epistasis motif to emerge (bottom), the fitness values need to be "interspersed".
(PDF)

**S3 Fig. Evolutionary consequences of landscape ruggedness in NK landscapes.** (A,C) The average length of a greedy adaptive walk and (B,D) the change in mean fitness at equilibrium, relative to fitness at equilibrium when selecting for $w_{opt} = 1$, are shown for (A,B) $L = 5$ and (C, D) $L = 8$. In (B,D), $\mu = 0.1$ and $\sigma = 0.5$.
(PDF)

**S4 Fig. Selection for $w_{opt} = 0$ can change the number and location of peaks.** Selection for $w_{opt} = 0$ does not change the type of epistasis motif for any "square" in the fitness landscape, relative to the genotype-phenotype landscape, yet it can change the number and location of peaks. To understand how, consider that any two adjacent faces of the hypercube (e.g., gray faces above) are sufficient to determine whether the genotypes on their common edge are peaks or not. After selecting for $w_{opt} = 0$, the type of epistasis motif does not change in the adjacent faces, yet the number and location of the peaks on their common edge does change. Peak genotypes are shown in red. Arrows point from lower to higher phenotypic or fitness values.
(PDF)

**S5 Fig. Sensitivity analysis for empirical landscapes.** Mean absolute change in the number of peaks in the fitness landscape relative to the genotype-phenotype landscape, shown in relation to $w_{opt}$, for (A) single-peaked vs. multi-peaked genotype-phenotype landscapes and (B) three values of $\sigma$.
(PDF)

**S6 Fig. Average peak height and width for empirical landscapes.** Average peak (A) height and (B) width for 1,137 empirical landscapes, shown in relation to the optimal binding affinity $w_{opt}$. Violin plots show the distribution across the landscapes for each $w_{opt}$. Data include both local and global peaks.
(PDF)

**S7 Fig. Greedy adaptive walks on NK landscapes.** The average height of peaks reached by greedy adaptive walks in NK landscapes with $\sigma = 0.5$, shown in relation to $w_{opt}$, for (A) $L = 5$ and (B) $L = 8$, with $K = 0 \ldots L - 1$.
(PDF)

**S8 Fig. Change in mean fitness at equilibrium for NK landscapes.** Change in mean fitness at equilibrium for NK landscapes with (A,B) L = 5 and (C,D) L = 8, shown in relation to $w_{opt}$, for different values of $\sigma$ and $\mu$.
(PDF)

**S9 Fig. Greedy adaptive walks on empirical landscapes.** (A) Local peaks on which the adaptive walks terminate tend to be nearly as tall as the global peaks in the 1,137 empirical landscapes. Violin plots show the distribution of the fractional height of local peaks reached by greedy adaptive walks, relative to the height of the global peak, for each optimal binding affinity $w_{opt}$. (B) Violin plots show the distribution of the fraction of walks terminating on the global peak, for each optimal binding affinity $w_{opt}$.
(PDF)

**S10 Fig. Change in mean fitness at equilibrium for mismatch model landscapes.** The mismatch class $m_{opt}$ that maximizes mean fitness at equilibrium is shown in relation to the strength of selection $\sigma$ and the mutation rate $\mu$. The three smaller panels show mean fitness at equilibrium in relation to $m_{opt}$ for three combinations of $\sigma$ and $\mu$. The value of $m_{opt}$ that maximizes mean fitness is indicated with a gray rectangle.
(PDF)

**S11 Fig. Sensitivity analysis for the change in mean fitness at equilibrium for mismatch model landscapes.** The mismatch class $m_{opt}$ that maximizes mean fitness at equilibrium is shown in relation to the strength of selection $\sigma$ and the mutation rate $\mu$ for three different values of the mismatch penalty $e$: (A) $e = 0.025$, (B) $e = 0.05$ and (C) $e = 0.1$.
(PDF)

**S12 Fig. Change in mean fitness at equilibrium for empirical landscapes.** The binding affinity $w_{opt}$ that maximizes mean fitness at equilibrium is shown in relation to the strength of selection $\sigma$ and the logarithm of the mutation rate ($\log \mu$) for the 1,137 empirical landscapes. The three smaller panels show the distributions of mean fitness at equilibrium as violin plots, in relation to $w_{opt}$, for three combinations of $\sigma$ and $\mu$. Box-and-whisker plots show the 25–75% quartiles. The value of $w_{opt}$ that maximizes mean fitness is indicated. These results are robust to perturbations of the fitness values (Methods), with the exception of two parameter combinations—$\sigma = 0.1$, $\mu = 0.001$ and $\sigma = 0.1$, $\mu = 0.0025$. For these parameter combinations, the binding affinity $w_{opt}$ that maximizes mean fitness at equilibrium changes from 0.35 to 0.385 (on average) and 0.375 respectively.
(PDF)

## Acknowledgments

We thank members of the Computational Biology Group at ETH, Michael Manhart and David M. McCandlish for discussions.

## Author Contributions

**Conceptualization:** Malvika Srivastava, Joshua L. Payne.

**Data curation:** Malvika Srivastava.

**Formal analysis:** Malvika Srivastava.

**Funding acquisition:** Joshua L. Payne.

**Investigation:** Malvika Srivastava, Joshua L. Payne.

**Methodology:** Malvika Srivastava.

**Project administration:** Joshua L. Payne.

**Resources:** Malvika Srivastava.

**Software:** Malvika Srivastava.

**Supervision:** Joshua L. Payne.

**Validation:** Malvika Srivastava.

**Visualization:** Malvika Srivastava.

**Writing – original draft:** Malvika Srivastava.

**Writing – review & editing:** Malvika Srivastava, Joshua L. Payne.

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
