## [Decision Letter · Decision Letter 0]

20 Jun 2022

Dear Prof. Payne,

Thank you very much for submitting your manuscript "On the incongruence of genotype-phenotype and fitness landscapes" for consideration at PLOS Computational Biology.

As with all papers reviewed by the journal, your manuscript was reviewed by members of the editorial board and by several independent reviewers. In light of the reviews (below this email), we would like to invite the resubmission of a significantly-revised version that takes into account the reviewers' comments.

While the reviewers found your work interesting, they made several suggestions for clarifications and raised additional questions. These points appear valid and should be addressed carefully, which will require extensive revisions. If you decide to submit a revised version, please respond to all reviewer comments, especially those of reviewer #2, and take them into account when preparing the revised manuscript.

We cannot make any decision about publication until we have seen the revised manuscript and your response to the reviewers' comments. Your revised manuscript is also likely to be sent to reviewers for further evaluation.

Sincerely,

Tobias Bollenbach

Associate Editor

PLOS Computational Biology

Ville Mustonen

Deputy Editor

PLOS Computational Biology

While the reviewers found your work interesting, they made several suggestions for clarifications and raised additional questions. These points appear valid and should be addressed carefully, which will require extensive revisions. If you decide to submit a revised version, please respond to all reviewer comments, especially those of reviewer #2, and take them into account when preparing the revised manuscript.

Reviewer's Responses to Questions

**Comments to the Authors:**

Reviewer #1: The manuscript “On the incongruence of genotype-phenotype and fitness landscapes” investigates how the ruggedness of fitness landscape is related to the underlying genotype-phenotype landscape when different selective pressures are applied. By using multisite biallelic genotype-phenotype landscapes and Gaussian phenotype-to-fitness map centered around phenotypic optimum, authors derive analytical relations for local and global properties of resulting fitness landscapes. The results are verified computationally for genotype-phenotype landscapes in the NK model and in the experimentally obtained dataset containing binding affinity of TF to DNA. The important finding is that selection for low and intermediate phenotypic values significantly increases ruggedness of the landscape, but interestingly the evolutionary accessibility of global peak is not affected.

The question of how different selective pressures modify both level of epistasis and number of adaptive peaks when genotype-phenotype landscape is mapped to the fitness landscape is a very interesting one with potential implications in both theoretical and experimental studies of biological evolution. The manuscript provides a very valuable contribution in terms of identifying different factors affecting local and global measures of ruggedness when mapping genotype-phenotype landscape into fitness landscape. I highly appreciate supporting computational results with analytical calculations that could guide our intuition about system behavior. The paper is well written, technically sound and results are well presented. My only major concern, is how the multi-allelic landscape is mapped to bi-allelic landscape, and whether this mapping is not biasing the evolutionary accessibility of resulting fitness landscape. Still, I expect this concern could be addressed. In summary, I find that this work deserves a broad dissemination and will be of interest for the PLoS Computational Biology audience.

Major issue

1. The multi-allelic genotype-phenotype landscape that mimics energy landscape of TF-DNA interactions is transformed into bi-allelic landscape through the threshold model. The threshold model results in genotype-phenotype landscape with large plateaus corresponding to low and intermediate phenotypic values. If now one phenotypic value corresponding to one of these plateaus is selected as most optimal it is not surprising that although the landscape is very rugged the optimal peaks can be reached with very short adaptive walks.

However, in more realistic approach the energy landscape of TF-DNA would likely not have these plateaus, but rather a spread of phenotypic values (see Mustonen et al. 2008, cited as [43]), with empirical distribution of these energies resembling rather a continuous bulk of non-specific bindings and additional peak for specific bindings. My objection with the current mismatch model is that applying a fitness map to the idealized plateau-like landscape, could be qualitatively different to applying the same fitness map to a landscape in which plateaus are not present. In practice one could introduce noise to phenotypic values in plateau-like landscape, or consider more fine-grained distinction of binding energies to introduce larger number of discrete phenotypic levels. The question would be whether fitness map will still result in fitness landscape with short adaptive walks toward the global fitness optimum?

I would appreciate at least a discussion of such case, as my impression is that current conclusion about no change in accessibility of adaptive peaks is simply a consequence of the plateau-like mapping introduced by threshold model.

Minor issues

2. lines 167-168, please give explicitly the cut-off for zero fitness, instead of “within computer precision”.

3. Eq 5 and lines 436-439, the properties of accessible pathways in multi-allelic landscapes were studied in Zagorski et al., PLoS Comp Bio 2016, indicating that increasing number of alleles would increase the average length of mutational pathways towards global optimum. Does it hold for eq. 5?

4. figure 7 caption, there are no “Closed symbols and dashed line” in figures 7B and 7D. Please remove or correct figures.

Reviewer #2: The topic of the paper are the changes in landscape properties that occur when a genotype-phenotype landscape is combined with a nonlinear phenotype-fitness map. Specifically, the authors consider a one-dimensional real-valued phenotype that is mapped to fitness via a nonlinear function with a unique optimal phenotype value. The primary motivation for the work is the perception that "genotype-phenotype landscapes are often used as a proxy for fitness landscapes", and that this may be potentially misleading. While I agree that this problem exists to some extent (mainly because organismal fitness is oftentimes a complex, multidimensional construct), I feel that it is also something of a straw man, since most workers in the field are very well aware of the distinction between the two concepts.

For the same reason, I'm not really happy with the title of the paper (though I know that it has already been modified

compared to the bioRxiv version of the manuscript). The authors address an important problem, viz., the transformation

properties of multidimensional phenotypic epistasis under nonlinear phenotype-fitness maps, and I think this could be better

reflected in the title. For the specific case where phenotypic epistasis is absent (and therefore the nonlinear phenotype-fitness

map is the sole source of epistasis for fitness) this problem has been addressed extensively (in the context of Fisher's geometric

model and elsewhere), and the authors cite a number of studies in this direction, but the generalization to epistatic

genotype-phenotype landscapes is novel and timely.

I am generally in favor of publication of the manuscript in PLoS Computationial Biology, but ask the authors to address the

following specific points prior to acceptance.

1. Related to the previous remark, I think the authors should emphasize that their analysis is not restricted to nonlinear

phenotype-fitness maps, but applies whenever one phenotype is mapped nonlinearly to another. In such a situation, the analysis of the resulting landscape can provide information about the underlying phenotypic architecture. As an example, Zwart et al. [Heredity 2018] concluded from an analysis of a genotype-resistance landscape for synonymous mutations in an antibiotic resistance enzyme that the underlying phenotype cannot be one-dimensional.

2. One of the most remarkable results of the paper is that simple sign epistasis motifs (SSE) cannot be transformed into

reciprocal sign epistasis (RSE) under a unimodal phenotype-fitness map. Unless I am mistaken, this has an even more remarkable (and quite counterintuitive) corollary. A well-known result by Poelwijk et al. (JTB 2011) states that a multipeaked fitness landscape has to contain at least one instance of RSE. Together with the new result, this would then imply that a genotype-phenotype landscape where all 2-faces have SSE remains single-peaked under arbitrary unimodal phenotype-fitness transformations [it is straightforward to see that such genotype-phenotype landscapes can be constructed for any L]. If the authors agree with this conclusion, I think it is worth mentioning.

3. Conceptually, it is not clear to me why the authors use the absolute change in the number of peaks as a measure of "incongruence", rather than just the different p_f - p_gp. Isn't is important to know whether the number of peaks increases or decreases?

4. Figure 4A and lines 225-227: I cannot see any maxima in the figure for L \\geq 4, and in fact I think the argument in the SM

(Note 1) is flawed. When the optimal phenotype is between two integer multiples of 1/L, there will be peaks in both layers of the cube and there is no reason why the total number of peaks should decline.

5. As it stands, Eq.(3) makes no sense, because the right hand side diverges with L. This should be rewritten to make it into a

proper limit statement.

6. Figure 7B: Where does the enormous variability of the adaptive walk length come from?

7. Supplementary figure 3: There are analytic results for the mean length of greedy adaptive walks in the HoC and NK-models

(e.g., Orr, JTB 2003; Nowak and Krug, J. Stat. Mech. 2015). In particular, the greedy walk length for the HoC model is

e-1 \\approx 1.7 for large L, which is smaller than the results shown in Fig. S3A and C. Do the authors possibly use a nonstandard definition of walk length where a 'zeroth' step is included?

**Have the authors made all data and (if applicable) computational code underlying the findings in their manuscript fully available?**

Reviewer #1: Yes

Reviewer #2: Yes
---

## [Decision Letter · Decision Letter 1]

30 Aug 2022

Dear Prof. Payne,

We are pleased to inform you that your manuscript 'On the incongruence of genotype-phenotype and fitness landscapes' has been provisionally accepted for publication in PLOS Computational Biology.

Best regards,

Tobias Bollenbach

Academic Editor

PLOS Computational Biology

Ville Mustonen

Section Editor

PLOS Computational Biology

Reviewer's Responses to Questions

**Comments to the Authors:**

Reviewer #1: In the revised manuscript authors have addressed all my major and minor concerns. In particular, authors performed simulations of landscapes with fine-grained distinction of phenotypic values, to validate whether accessibility of such landscapes is affected in a different way than in the landscapes with plateau regions. As a result, authors find that there are quantitative differences in the difference of peaks, the length of adaptive walks, and fraction of walks that reach global peak, but the overall trends are qualitatively the same for fine-grained and plateau-like landscapes. This was illustrated in the reviewer figures and addressed in the text of the revised manuscript.

I find the revised manuscript to provide a very valuable contribution to systems and computational biology communities and strongly support its publication in PLoS Computational Biology.

Reviewer #2: The authors have fully addressed the comments in my previous report. I recommend publication of the manuscript in its present form.

**Have the authors made all data and (if applicable) computational code underlying the findings in their manuscript fully available?**

Reviewer #1: Yes

Reviewer #2: Yes

PLOS authors have the option to publish the peer review history of their article (what does this mean?). If published, this will include your full peer review and any attached files.

Reviewer #1: No

Reviewer #2: **Yes: **Joachim Krug

---

## [Editor Report · Acceptance letter]

13 Sep 2022

PCOMPBIOL-D-22-00693R1 

On the incongruence of genotype-phenotype and fitness landscapes

Dear Dr Payne,

I am pleased to inform you that your manuscript has been formally accepted for publication in PLOS Computational Biology. Your manuscript is now with our production department and you will be notified of the publication date in due course.

With kind regards,

Zsofi Zombor
